# Kingdom-wide comparison reveals the evolution of diurnal gene expression in Archaeplastida

Camilla Ferrari[1], Sebastian Proost[1], Marcin Janowski[1], Jörg Becker [2], Zoran Nikoloski[1,3], Debashish Bhattacharya[4], Dana Price[5], Takayuki Tohge[1,6], Arren Bar-Even [1], Alisdair Fernie[1], Mark Stitt[1] & Marek Mutwil[1,7]

Plants have adapted to the diurnal light-dark cycle by establishing elaborate transcriptional programs that coordinate many metabolic, physiological, and developmental responses to the external environment. These transcriptional programs have been studied in only a few species, and their function and conservation across algae and plants is currently unknown. We performed a comparative transcriptome analysis of the diurnal cycle of nine members of Archaeplastida, and we observed that, despite large phylogenetic distances and dramatic differences in morphology and lifestyle, diurnal transcriptional programs of these organisms are similar. Expression of genes related to cell division and the majority of biological pathways depends on the time of day in unicellular algae but we did not observe such patterns at the tissue level in multicellular land plants. Hence, our study provides evidence for the universality of diurnal gene expression and elucidates its evolutionary history among different photosynthetic eukaryotes.

[1] Max-Planck Institute for Molecular Plant Physiology, Am Muehlenberg 1, 14476 Potsdam, Germany. [2] Instituto Gulbenkian de Ciência, R. Q.ta Grande 6, 2780-156 Oeiras, Portugal. [3] Bioinformatics Group, Institute of Biochemistry and Biology, University of Potsdam, Karl-Liebknecht-Str. 24-25, 14476 Potsdam, Germany. [4] Department of Biochemistry and Microbiology, Rutgers University, New Brunswick, NJ 08901, USA. [5] Department of Plant Biology, Rutgers University, New Brunswick, NJ 08901, USA. [6] Graduate School of Biological Sciences, Nara Institute of Science and Technology, Ikoma, Nara 630-0192, Japan. [7] School of Biological Sciences, Nanyang Technological University, 60 Nanyang Drive, Singapore 637551, Singapore. Correspondence and requests for materials should be addressed to M.M. (email: mutwil@ntu.edu.sg)

The endosymbiotic event that occurred between a eukaryotic ancestor and cyanobacteria led to the formation of the plastid and emergence of early single-celled members of the Archaeplastida. These organisms established multicellularity, conquered land, and evolved complex organs to adapt and survive the new environmental challenges. Comparative genomic studies aim to reveal how changes in genomes and gene families have facilitated the evolution of new cellular and physiological features[1,2] and have led to important discoveries. These include the single origin of the plastid in eukaryotes[3] and the emergence and expansion of gene families that are essential for drought tolerance and for hormone synthesis during the appearance of land plants[4]. However, on its own, genome sequence data might not reveal which genes work together towards establishing a trait, and how new traits are formed from new or existing genes[5–7]. To address this paucity of functional information from genome data, researchers compare other gene functional data, such as protein–protein interactions[8] and gene-expression (transcriptomic) data[9]. Comparative transcriptomics were used to identify molecular mechanisms underlying interspecific differences related to plant adaptation[10] and to identify the evolution of metabolic responses in microalgae[11]. Furthermore, comparative transcriptomics can be used in a broader fashion to discover shared and unique events that occurred in evolution. This can be accomplished by comparing distantly related species, as exemplified by the identification of modules that are involved in development and that are shared across human, worm, and fly[12] and by the discovery of conserved early and late phases of development in vertebrates[13].

Photosynthetic organisms are capable of perceiving day length and the alternation of light and dark through two primary systems: light–dark detection and the circadian clock[14]. In addition, light acts as the energy source that drives photosynthesis. The circadian clock, light signaling, and the supply of energy and fixed carbon are the driving forces of metabolism and of many biological processes whose gene expression is regulated by alternating light and dark intervals. Diurnal regulation actively controls physiological processes, such as cell division, metabolic activity, and growth in all domains of life[15–18]. Gene expression plays a fundamental role in controlling the activity of biological pathways and is under strict diurnal regulation. For example, in *Ostreococcus tauri* most of the genes are under control of light/dark cycles and specifically, genes involved in cell division, the Krebs cycle and protein synthesis, have the highest amplitude within the diurnal cycle[18,19]. Studies in *Drosophila melanogaster* show that in addition to gene expression, splicing, RNA editing, and noncoding RNA expression are highly affected by diurnal regulation[20], whereas studies in mammals highlight the importance of diurnal regulation when applied to pharmaceutical applications[21,22]. Furthermore, recent studies on plants have shown a moderate level of conservation of diurnal responses among distantly related species, along with occasional divergent evolution of specific mechanisms[14,18].

In this study, we analyzed diurnal transcriptomes of distantly related species of Archaeplastida, including eukaryotic algae, different phylogenetic groups of terrestrial plants, and a cyanobacterium as an outgroup. Our overarching goal was to understand how diurnal gene expression has evolved to accommodate the appearance of new gene families, morphologies, and lifestyles. To this end, we generated, collected and compared diurnal gene-expression data of model organisms representing nine major clades of photosynthetic eukaryotes, comprising *Cyanophora paradoxa* (glaucophyte, early diverging alga), *Porphyridium purpureum* (rhodophyte), *Chlamydomonas reinhardtii* (chlorophyte), *Klebsormidium nitens* (charophyte), *Physcomitrella patens* (bryophyte), *Selaginella moellendorffii* (lycophyte), *Picea abies*

(gymnosperm), *Oryza sativa* (monocot), and *Arabidopsis thaliana* (eudicot). We demonstrate that diurnal transcriptomes are significantly similar (false-discovery rate (FDR) corrected empirical $p$ value < 0.05) despite the large evolutionary distances and differences in morphological complexity and habitat among the sampled taxa. As previously shown, we find that cell division of the single-cellular and simple multicellular algae is synchronized by light. Conversely, land plants show a more uniform expression of the cell division genes, suggesting that cell division is either not taking place in most of the sampled cell types, or is not under diurnal control. Establishment of multicellularity has also resulted in a broader phase of peak expression of genes involved in most biological processes which, could be caused by a less defined diurnal control, or conversely, by a more elaborate diurnal regulatory control found in the different tissue types. Finally, we show that the core components of the circadian clock show conserved expression among plant clades, thus suggesting a potential mechanism behind conserved diurnal gene expression. Our study represents a significant advance in our understanding of how diurnal gene expression evolved in the anciently derived Archaeplastida.

## Results

**Diurnally regulated genes in Archaeplastida.** To generate comprehensive diurnal expression atlases capturing the evolution of the Archaeplastida, we retrieved publicly available diurnal gene-expression data from *Synechocystis* sp. PCC 6803[23], *C. reinhardtii*[17], *O. sativa* seedlings[24], and *A. thaliana* rosettes[25], and generated new RNA sequencing data from *C. paradoxa*, *P. purpureum*, *K. nitens*, *P. patens* gametophores, *S. moellendorffii* aerial tissues, and *P. abies* seedling needles (Fig. 1a). To generate the data, we subjected *P. purpureum*, *K. nitens*, *P. patens*, *S. moellendorffii* to a 12-h light/12-h dark photoperiod (12L/12D) and *C. paradoxa*, *P. patens*, and *P. abies* to a 16L/8D photoperiod. Different photoperiods were used because for species such as *P. abies* (herein referred to as spruce), a shorter photoperiod would cause dormancy and growth cessation[26]. The difference in photoperiod was shown to have a relatively small impact on the organization of daily rhythms[18]. Starting from ZT1 (Zeitgeber time), defined as 1 h after the onset of illumination, we extracted RNA from the 6 organisms every 2 h, resulting in 12 time points per species. We obtained an average of 20 million reads per sample (Supplementary Data 1), whereby more than 85% of the reads mapped to the genome, and on average 70% of the reads mapped to the coding sequences. Principal component analysis (PCA) of the samples showed separation according to the time and between light and dark samples for most of the species, with exception of spruce (Supplementary Fig. 1), suggesting that gene expression in spruce is less controlled by light.

To detect genes that are regulated by the diurnal cycle and to elucidate the time of the day where these genes reach peak expression, we set out to identify genes showing rhythmic gene expression. To this end, we used the JTK_Cycle algorithm[20] which allowed us to determine significantly rhythmic genes (adjusted $p$ value < 0.05) and the phase of peak expression of these genes. The phases range from 1 (the gene peaks at the first time-point of the series) to 24 (the gene peaks at the last time-point, Supplementary Datas 2–11).

We observed that between 5.2 and 55.9% of genes show significantly rhythmic expression across species (adjusted $p$ value < 0.05, Fig. 1b). We observed that single-celled algae tend to be on average more rhythmic (mean of 49.3% rhythmic genes over *C. paradoxa*, *P. purpureum*, and *C. reinhardtii*), than multicellular land plants (mean of 31.8% rhythmic genes over *P. patens*, *S. moellendorffii*, *O. sativa*, and *A. thaliana*). We

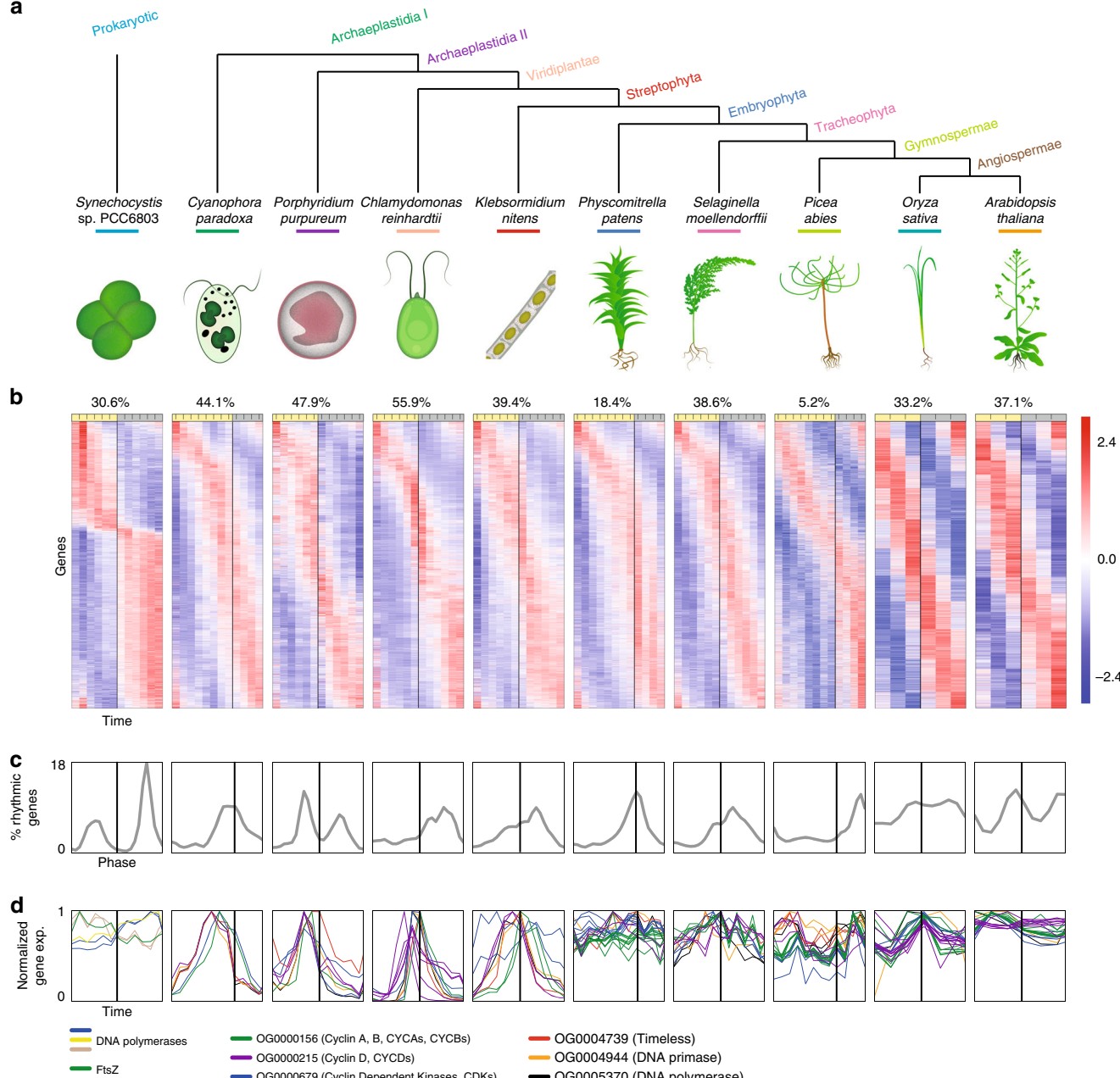

**Fig. 1** Diurnal gene expression of plant clades. **a** Species tree of the Archaeplastida, together with the respective phylostratum of each clade. *Synechocystis* sp. PCC 6803 is included as a cyanobacterial outgroup. **b** The percentage of rhythmic genes in a given species is shown above the yellow (light) and gray (darkness) bar indicating the photoperiod of each diurnal experiment. Below, z-score-standardized gene-expression profiles for each species sorted by the phase, signifying the time point of highest expression during the day. **c** Phase distribution of rhythmic genes through the diurnal experiment. **d** Expression profiles of cell division genes. The coloured lines indicate the different gene families. Each family can have more than one gene

excluded spruce from this calculation, as it showed an unusually low percentage of rhythmic genes (5.2%). However, this is in line with studies on diurnal expression of genes involved in photosynthesis, which are largely rhythmic in flowering plants, but not in spruce and other gymnosperms[27–29].

To elucidate when rhythmic genes peak during the day, we plotted the number of genes assigned to each phase (Fig. 1c). From glaucophyte to gymnosperm, the majority of rhythmic genes tend to peak during the second half of the day, at either dusk or night. The exception to this trend is the red alga *P. purpureum*, in which rhythmic genes have a bimodal distribution, in the middle of the light and dark period, similarly to the cyanobacterium *Synechocystis* sp. PCC 6803. Furthermore, in

angiosperms, we observed a more uniform distribution of rhythmic genes with a bimodal distribution coinciding with transitions from day to night and from night to day (Fig. 1c). The more uniform expression suggests that, in contrast to algae and early diverging land plants, gene expression in angiosperms is less under the control of the diurnal cycle. Alternatively, the uniform diurnal gene expression in angiosperms suggests a more nuanced regulatory control allowing access to more phase bins.

The diurnal cycle is known to synchronize cell division in cyanobacteria[30], red algae[31], and green algae[32], and cell division is accompanied by a sharp increase in the transcript levels of DNA polymerases, primases, cyclins, and other proteins[17,19]. The expression of these marker genes revealed that single-celled algae

and the monotypic multicellular *K. nitens* exhibit a single expression peak of cell division genes at the onset of darkness (Fig. 1d). Conversely, the more complex multicellular land plants show, at the tissue level, uniform expression of these genes throughout the day (Fig. 1d). This suggests that either the increased morphological complexity of multicellular land plants necessitated uncoupling of the diurnal cycle and cell division, or that most cells in multicellular plants are not actively dividing, regardless of the diurnal cycle. The latter hypothesis is supported by the observation that the circadian clock controls cell division in actively dividing cells of *A. thaliana* seedlings[33]. Future studies of cell-specific transcriptomics data would facilitate the understanding of cell division regulation in complex multicellular organisms.

**Phylostrata analysis of rhythmicity and expression**. The age of appearance of gene families is positively correlated with their expression levels and severity of the corresponding mutant phenotypes[5]. To investigate whether there is a relationship between the age of a gene family and its diurnal expression pattern, we first defined the age of gene families with a phylostratigraphic analysis, which identifies the earliest clade found within a gene family[1]. Because the order of appearance of the clades is known (Fig. 1a), gene families can be sorted according to their relative age (phylostrata), from oldest phylostratum (i.e., present in cyanobacteria and all Archaeplastida) to youngest (e.g., specific to only *A. thaliana* or *O. sativa*, Supplementary Data 12). This analysis revealed that the majority of genes belong to the earliest phylostratum for algae and plants (Fig. 2a, green bar), suggesting that most genes in plants have ancient origin. Next, we plotted the distribution of expression peaks of the rhythmic genes from the different phylostrata, and observed similar diurnal expression patterns for all phylostrata (Fig. 2b, an example from *P. patens*, Supplementary Fig. 2 for other species), indicating that the age of genes does not strongly influence the diurnal timing of gene expression.

We also explored if there is a relationship between the age of gene families and their rhythmicity, by comparing the percentage of significantly rhythmic genes within each phylostratum. There was a significant tendency for older phylostrata to have a higher proportion of rhythmic genes than younger phylostrata (Fig. 2c, FDR corrected empirical *p* value). For example, the oldest phylostratum, Prokaryotic (containing genes found in bacteria and eukaryotes, light blue bar), was more rhythmic than expected by chance for most of the analyzed organisms (Prokaryotic and Archaeplastida I phylostrata, FDR corrected empirical *p* value < 0.05). Conversely, the youngest phylostrata, comprising species-specific or clade-specific genes (specific phylostratum, gray bar, FDR corrected empirical *p* value < 0.05) were less rhythmic than expected by chance for all species, with the exception of *C. reinhardtii* and spruce. Since rhythmicity tends to be retained after gene duplication[34], we hypothesize that the high rhythmicity of the old phylostrata has ancient origin.

Finally, we asked whether the age of a gene family influences the expression levels of the transcripts, and observed that the gene-expression levels significantly decrease as the phylostrata become younger, and the lowest expression is reported for species-specific or clade-specific gene families (Fig. 2d, Kolmogorov–Smirnov (K–S) test, FDR corrected *p* value < 0.05, Supplementary Fig. 3). Since we have observed these trends in single-cellular organisms, we propose that genes belonging to older phylostrata are more strongly expressed in all cell types.

**Diurnal transcriptomes are similar**. To determine if the diurnal transcriptomes of the studied species are similar, we investigated whether orthologous genes peak at a similar time during the day in the ten species. To this end, we first identified orthologs among the species through protein sequence similarity analysis using OrthoFinder[35]. To avoid analyzing unclear orthologous relationships, which can arise by gene duplications, we only considered orthologs which showed a one-to-one relationship.

We visualized similarities across two transcriptomes with a heatmap, which indicates when the orthologs in two species peak during the day (Fig. 3a, an example from the *C. reinhardtii*–*C. paradoxa* comparison, Supplementary Data 13). The heatmap would show two similar transcriptomes as points found in a diagonal that starts in the upper-left corner and ends in the lower-right corner (see an example of similar transcriptomes in Supplementary Fig. 4). The comparison between *C. reinhardtii* (12L/12D) and *C. paradoxa* (16L/8D) shows that, despite the four hour difference in the photoperiod in which the algae were grown, most of the orthologs of these two species tend to be found on the diagonal of the plot, indicating that the orthologs of these two species peak at a similar time of the day.

To estimate if diurnal transcriptome time-series are significantly similar in two species, we calculated the phase differences ($\Delta$phase) (i.e., differences in peaking times) between the orthologs (Fig. 3b, dark blue bars), expecting that the average phase differences of significantly similar transcriptomes would be smaller than expected by chance. To this end, we compared the observed phase differences ($\Delta$phase$_{observed}$) with the phase differences calculated from permuted expression data ($\Delta$phase$_{expected}$, Fig. 3b, light blue bars). Indeed, the analysis revealed that, in contrast to evenly distributed $\Delta$phase$_{expected}$ values, the $\Delta$phase$_{observed}$ values peaked at 2–3 h (Fig. 3b). This indicates that *C. paradoxa* reaches the highest similarity to *C. reinhardtii* when its phases are offset by +2 h. Furthermore, comparison of the $\Delta$phase$_{observed}$ and the $\Delta$phase$_{expected}$ values revealed that the diurnal transcriptomes of *C. reinhardtii* and *C. paradoxa* are significantly similar (FDR corrected empirical *p* value < 0.012, obtained from 1000 permutations of expression data).

Interestingly, because $\Delta$phase$_{observed}$ values peak at 2–3 h when comparing *C. paradoxa* and *C. reinhardtii*, *C. paradoxa* genes tend to peak 2 h earlier than the corresponding *C. reinhardtii* genes. Consequently, the two transcriptomes would be more similar (i.e., average $\Delta$phase$_{observed}$ closer to 0) if the phase of *C. paradoxa* genes were offset by 2 h later in the diurnal cycle. To test this, we calculated $\Delta$phase$_{expected}$/$\Delta$phase$_{observed}$ value between *C. paradoxa* and *C. reinhardtii*, upon shifting the phase of *C. paradoxa* genes by an integer in the interval of [−12, 12]. The value becomes larger than one in cases where the observed phase difference ($\Delta$phase$_{observed}$) is smaller (i.e., more similar) than the phase differences obtained from the permuted expression data ($\Delta$phase$_{expected}$). Indeed, we observed highest $\Delta$phase$_{expected}$/$\Delta$phase$_{observed}$ value when a shift of +2 is applied to *C. paradoxa* transcriptome (Fig. 3c). Accordingly, the diagonal pattern in the heatmap became more evident after the shift (Fig. 3d, Supplementary Data 13), which was accompanied by a more significant *p* value (Supplementary Fig. 5). Thus, we conclude that the diurnal transcriptomes of *C. paradoxa* and *C. reinhardtii* are significantly similar, but there is on average, a 2 h difference between when the orthologs in these two organisms peak.

To investigate similarities of the transcriptomes across the Archaeplastida, we applied this analysis for all possible species combinations (Fig. 3e), and we observed that several comparisons show significant similarity even without applying a shift (solid thick frame, FDR corrected empirical *p* value < 0.05) and most comparisons reach a significant similarity when the shift to their phases is applied (dashed thick frame, FDR corrected empirical *p* value < 0.05; see Supplementary Fig. 5 upper-left triangle for the

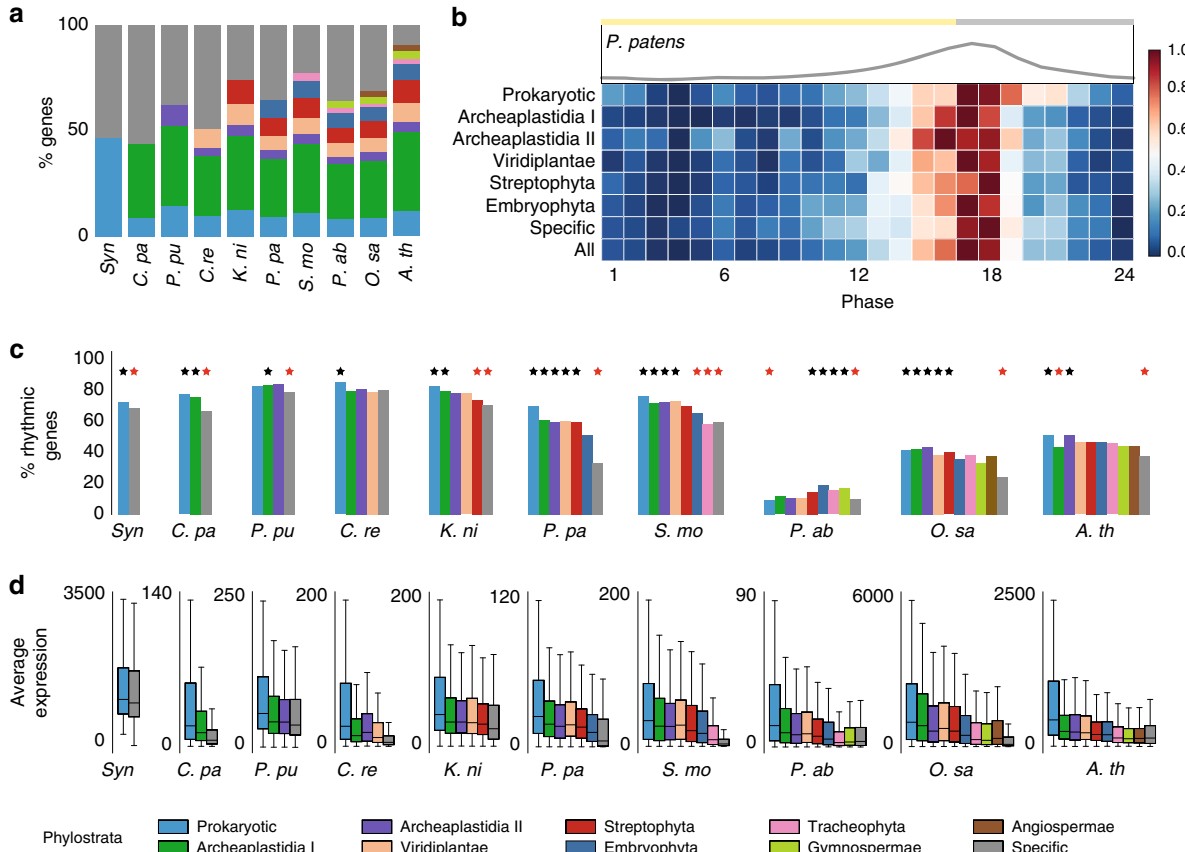

**Fig. 2** Expression analysis of phylostrata. **a** Percentages of genes in each phylostrata per species. **b** Phase distribution for the phylostrata in *P. patens*. The plot above the heatmap shows the cumulative distribution of the phases. **c** Analysis of phylostrata rhythmicity per species. The bars represent the percentage of rhythmic genes for each phylostratum. Black and red stars indicate phylostrata that are significantly enriched or depleted for rhythmic genes, respectively (FDR corrected empirical *p* value < 0.05). **d** Average expression of genes assigned to the phylostrata in the different species. The center line and the bounds of the box represent the median, the first and third quartile, respectively. The whiskers represent the rest of the distribution

shift applied in each case). Spruce is an exception, being similar to only five species. Furthermore, spruce produces the most extreme shift values required to obtain significantly similar transcriptomes, as exemplified by comparison with *S. moellendorffii* (shift of 5 h), *P. patens* (6 h), and *P. purpureum* (9 h, Supplementary Fig. 5).

To understand which biological processes show conserved rhythmic expression across Archaeplastidia, we calculated the fraction of the orthologs that were rhythmic. We observed a conservation of rhythmicity of signaling, tetrapyrrole synthesis, cell division, protein, RNA, and DNA synthesis and processing pathways (Supplementary Fig. 6). Conversely, gluconeogenesis, oxidative pentose pathway, TCA cycle and amino acid metabolism were among the less rhythmic pathways, suggesting that diurnal regulation of primary metabolic pathways is less conserved (Supplementary Fig. 6).

To better understand how developmental stages and the photoperiod variation influences the observed rhythmicity, we compared diurnal data from *A. thaliana* seedlings (12L/12D, 22 °C day/12 °C night and constant 22 °C, in 16L/8D and 8L/16D)[36] and rosettes (constant temperature, 12L/12D)[25] (Supplementary Fig. 7A). We observed that rosettes contain a higher percentage of rhythmic genes (37%) than seedlings in any of the seedling experiments (10.2, 17, and 26.4%, Supplementary Fig. 7A). This was also observed in *O. sativa*, where seedlings (33.2%) showed lower percentage of rhythmic genes than flag leaves (43.5%) (Supplementary Fig. 7B)[24]. While the diagonal distribution of the shift values indicated that the diurnal transcriptomes are highly

similar across the photoperiods, developmental stages and laboratories (Supplemental Fig. 7A, B), we observed a substantial average difference between phases of *A. thaliana* rosettes and seedlings (rosettes peak 2.83, 2.98 and 4.96 h earlier than 12L/12D, 16L/8D, and 8L/16D seedlings). Conversely, we observed only a modest difference of 0.5 h between seedlings and flag leaves of *O. sativa*, suggesting that the developmental stage has a minor influence on diurnal gene expression in rice (Supplemental Fig. 7B). These results indicate that while the genes tend to robustly peak in the same sequence across different conditions, the overall phase of the genes can be substantially influenced by the environment and developmental conditions.

**Diurnal gene expression of biological processes**. To further compare the diurnal gene expression of the Archaeplastida, we investigated the expression of genes related to the major cellular pathways of every species. To this end, we analyzed the expression profiles and the average expression of genes assigned to MapMan ontology bins[37]. The analysis revealed that specific organisms show stronger rhythmicity for specific processes.

For example, most of the photosynthesis-related genes are rhythmic in *C. paradoxa*, with highest expression during the light period, with a decrease below 50% of maximum expression during the second half of the light period (Fig. 4a). Conversely, the corresponding orthologs in spruce are not rhythmic and expressed within 80% of maximum levels at most time points of the day (Fig. 4b, dashed squares). The average expression of the

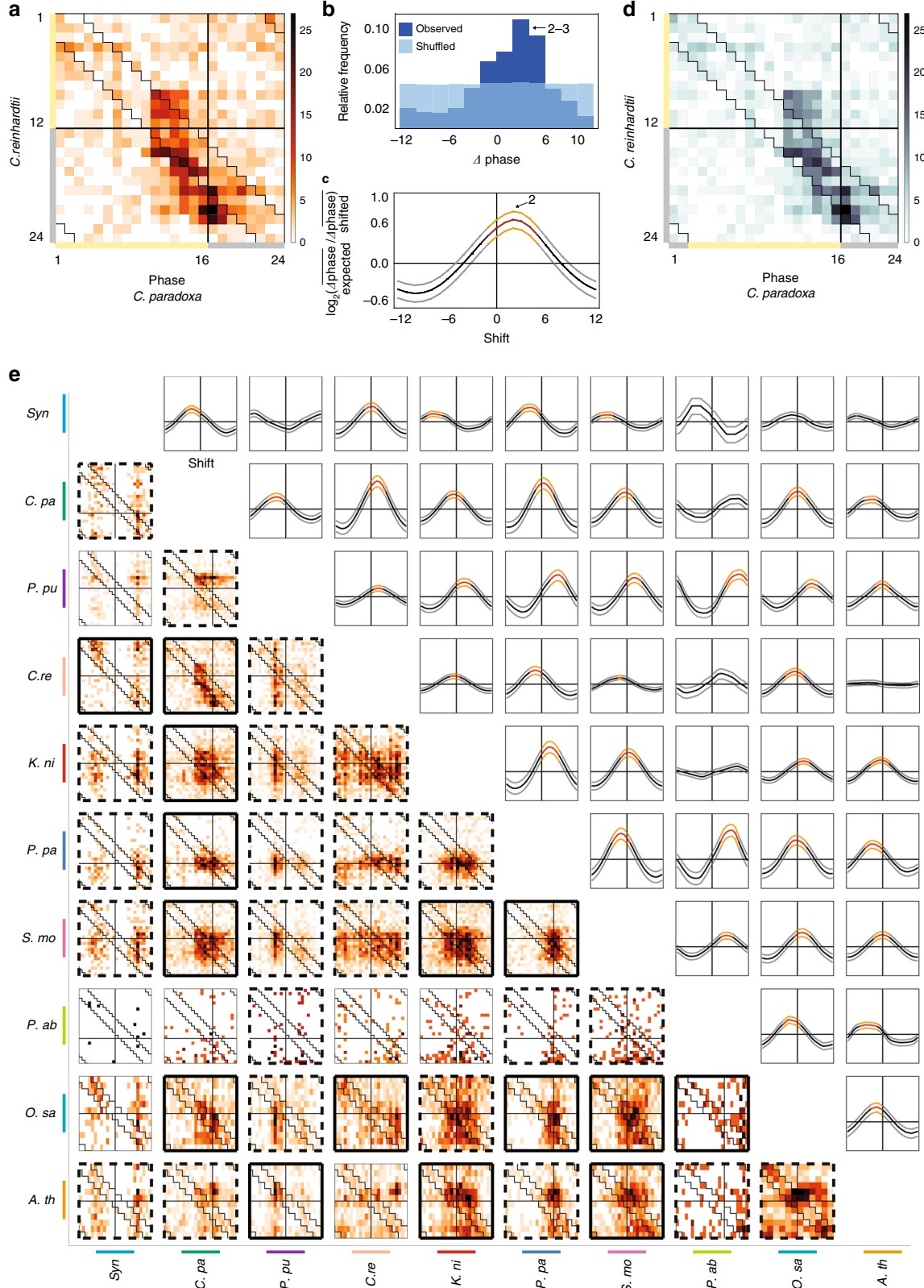

photosynthesis genes is summarized by the "photosynthesis" heatmap which showed that photosynthesis genes are on average above 80% of their maximum expression levels in three and 10 out of 12 time points for *C. paradoxa* and spruce, respectively (Fig. 4a, b, dashed squares). This result indicates that in spruce, in contrast to *C. paradoxa*, the expression of photosynthesis genes is not light-circadian dependent.

To search for similarities and differences in diurnal expression of biological processes in Archaeplastida at a more general level, we performed this analysis for all MapMan bins (Fig. 4c). In line

**Fig. 3** Comparison of diurnal transcriptomes. **a** Phase comparison of orthologs of *C. reinhardtii* (*y*-axis) and *C. paradoxa* (*x*-axis). The color intensity of the cells indicates the number of orthologs that peak at a given phase combination in the two species. Black lines indicate a transition from light to dark, while the zigzag lines indicate a phase differences within ±Δ2 h between the two species. **b** Relative frequency of phase differences between orthologs (Δphase$_{observed}$, dark blue bars, Δphase$_{expected}$, light blue bars) in *C. reinhardtii* and *C. paradoxa*. Δphase values are binned in twos, where the first bar indicates [−12, −11], the second bar [−10, −9], and so on. **c** Influence of the shift on transcriptome similarity. The middle thick line indicates the median of the log$_2$(Δphase$_{expected}$/Δphase$_{shifted}$) values, whereas the upper and lower lines indicate the third and first quartiles, for every shift of *C. paradoxa* phase values within the interval of [−12, 12]. The colors indicate significantly (empirical *p* value < 0.05) similar (orange, red) and not significantly similar (gray, black) transcriptomes upon the shift. **d** Phase comparison of orthologs after the shift of +2 h is applied to *C. paradoxa*. **e** Comparison of all possible species combinations (heatmaps and line plots). The thick black frames of the heatmaps indicate which species combination has a significantly different Δphase$_{observed}$ (FDR corrected empirical *p* value < 0.05) from the Δphase$_{expected}$, the dashed thick frames indicate which Δphase$_{observed}$ is significantly different (FDR corrected empirical *p* value < 0.05) than the Δphase$_{expected}$ after the shift. The thin frames indicate combinations whose distribution is not significantly different than the permuted distribution

---

with the observation that cell division is not similarly controlled in land plants (Fig. 1d), genes belonging to "cell" bin (which contains genes involved in cell division) in multicellular organisms have a more uniform expression throughout the day, compared to the defined expression of unicellular plants (Fig. 4c, uniformly expressed genes indicated by dark green color). Furthermore, we observed that biological processes in single-celled algae, such as *C. paradoxa*, *P. purpureum*, and *C. reinhardtii* tend to be expressed at specific times during the day (Fig. 4c, genes belonging to specifically expressed processes are indicated by light green cells). Conversely, in the relatively simple multicellular *K. nitens* and in land plants, biological processes show a more uniform pattern of diurnal gene expression (Fig. 4c, dark green cells). This suggests that establishment of multicellularity, rather than colonization of land, decreased the diurnal regulation of gene expression.

Finally, we examined how the rhythmicity of biological processes differs between species (Fig. 4d). We observed that in single-celled organisms, basal metabolic processes, such as the oxidative pentose pathway, major carbohydrate metabolism, fermentation, tetrapyrrole synthesis, biodegradation of xenobiotics, photosynthesis, glycolysis, and tricarboxylic acid (TCA) cycle are the most rhythmic, with between 60 and 100% of the genes assigned to the processes showing significant rhythmic behavior. Conversely, we observed a decrease of rhythmicity of most biological processes in *K. nitens* and land plants, with spruce showing less than 20% of rhythmic genes for all the investigated processes.

**Analysis of circadian clock components**. The circadian clock has been extensively studied in *A. thaliana*[38,39] and it is known to regulate diurnal expression of thousands of transcripts[15,38] to coordinate diverse biological processes so that they occur at the most appropriate season or time of day. The core of the clock is composed of three integrated feedback loops (Fig. 5a) whose components are fine-tuned by additional regulatory factors. A morning loop where *CIRCADIAN CLOCK ASSOCIATED 1* (*CCA1*) and *LATE ELONGATED HYPOCOTYL* (*LHY*) together positively regulate the expression of *PSEUDO RESPONSE REGULATOR* (*PRR*) *7* and *9*, while negatively regulate *PRR5* gene, whereas PRRs in response, negatively regulate the two genes[40]. It is important to note that the positive regulation of *PRR7* and *9* by *CCA1/LHY* is probably indirect, as *CCA1/LHY* are shown to be repressors[41], suggesting that these two genes are involved in a yet uncharacterized regulatory circuit. The central loop has as protagonists *CCA1/LHY* and *TIME OF CAB EXPRESSION 1* (*TOC1*, also known as *PRR1*) which negatively regulate one another[41]. The last loop includes the evening complex, activated by the *REVEILLE* genes, formed by *LUX ARRHYTHMO* (*LUX*), *EARLY FLOWERING 3* and *4* (*ELF3*, *ELF4*), which represses *TOC1* and

the other members of *PRR*, together with *GIGANTEA* (*GI*) and *ZEITLUPE* (*ZTL*) involved in the degradation of the proteins, ensuring the expression of the morning-phased components *CCA1* and *LHY*[42–44].

To explore how the clock as observed in Arabidopsis evolved, we analyzed the expression of eight gene families containing the core clock genes from *A. thaliana* (*CCA1*, *LHY*, *RVE1–8*, *PRR3*, *5*, *7*, *9*, *TOC1*, *LUX*, *ELF3*, *ELF4*, *GI*, and *ZTL*) in the nine members of the Archaeplastida (Fig. 5c–i). Furthermore, we have included *O. tauri*, as the diurnal gene expression of this chlorophyte has been well studied[19]. As previously shown[14], only *CCA1/LHY/RVEs* and *LUX* families are present in most of the species, indicating that the three major loops are not complete in the early diverging algae *C. paradoxa* and *P. purpureum*, because *PRR* genes (morning loop), *TOC1* (central loop), *ELF3/ELF4/GI/ZTL* (evening loop) are absent in these two algae (Fig. 5a, c–i). These results suggest that despite the lack of conservation of the Viridiplantae clock, conserved diurnal gene expression can be mediated by other mechanisms, such as light sensor-mediated response to light[14,18], carbon signaling[45], an uncharacterized or a modified type of clock where the known components have been reconfigured to operate with other transcription factors involved in clock regulation[46,47]. Furthermore, the different loops were established at specific times in plant evolution, because genes constituting the morning and central loop are first found in the Viridiplantae clade (containing *C. reinhardtii*, *O. tauri*, *K. nitens* and land plants, Supplementary Fig. 9), whereas a complete evening loop was established in streptophytes (containing *K. nitens* and land plants, Supplementary Figs. 10–12).

Next, we compared expression patterns of the clock genes (Fig. 5c–i). The *PRR* genes show conserved expression during the day across all species (Fig. 5b, d, Supplementary Fig. 9), with the exception of *C. reinhardtii* and *O. tauri* PRRs, which are mostly expressed during the night. In fact, whereas all species showed the highest expression of the genes at dusk, *C. reinhardtii* and *O. tauri* PRRs peaked at night, confirming previous analyses[18,48]. The larger gene families exhibited more complex expression patterns. Genes in *CCA1/LHY/RVEs* were expressed predominantly at the end of night/beginning of the day in all the species (Fig. 5c), whereas *ELF3* showed a different expression at the end of the day/beginning of the night (Fig. 5h).

Our analyses reveal that spruce has the smallest fraction of rhythmic genes, with only 5.2% genes showing significantly rhythmic behavior (Fig. 1b). Surprisingly, all of the clock components are present in spruce (Fig. 5c–i), with the exception of *ELF3*, suggesting that the spruce clock might be more active at different developmental stages, temperature oscillations or photoperiods than were used in this study. Indeed, shorter photoperiods in spruce (8L/16D) are accompanied by an increase of amplitude of clock genes[27]. We observed that *GI* (Fig. 5g) and

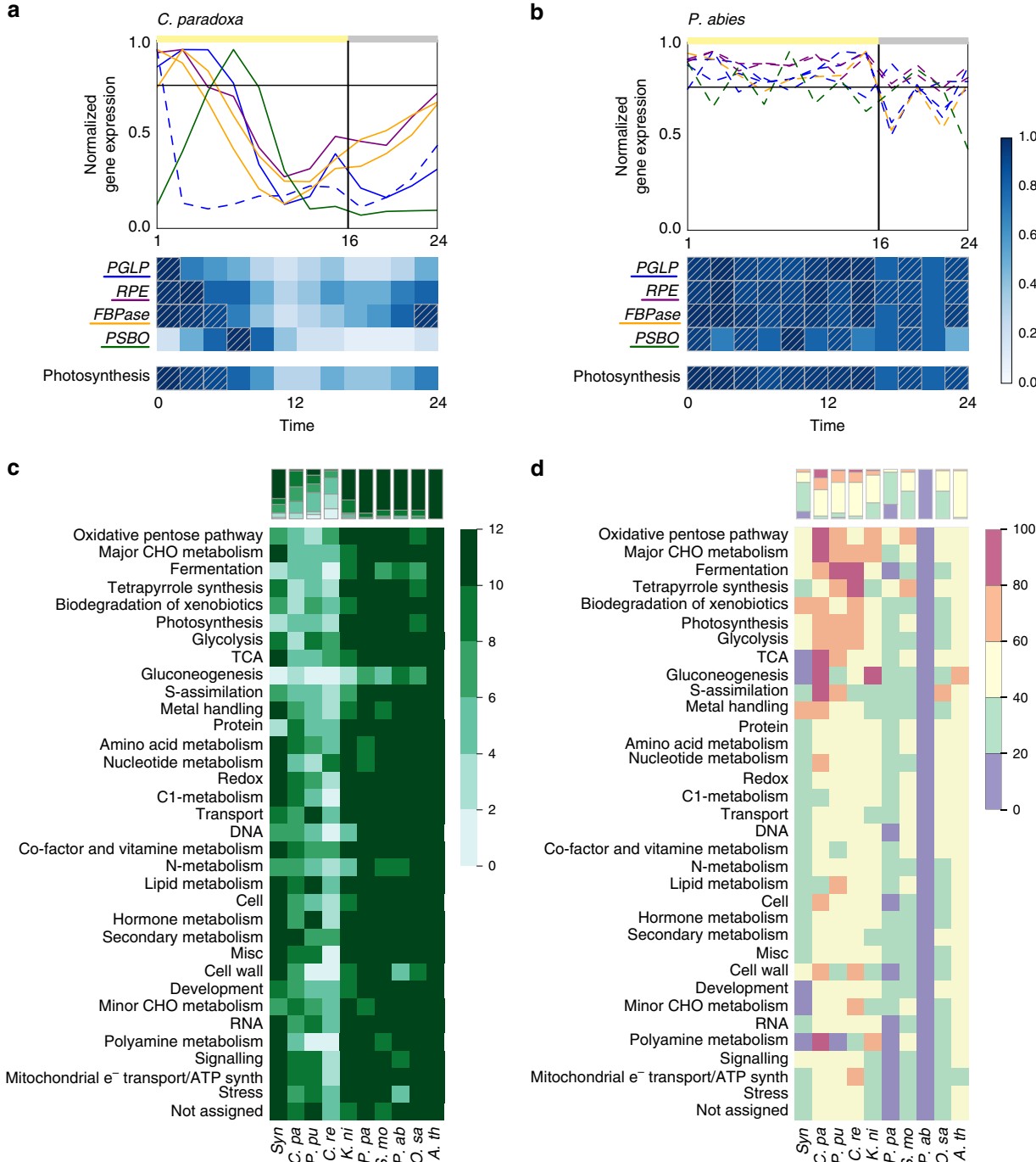

**Fig. 4** Diurnal expression of biological processes. **a** Gene-expression profiles of *C. paradoxa* genes belonging to four families involved in photosynthesis. Solid lines indicate significantly rhythmic, while dashed lines indicate arrhythmic genes. The horizontal black line indicates the 80% expression threshold. The heatmap below the gene expression plot shows the average expression of the four gene families. Dashed squares indicate the average expression values higher than 80% at a given time point. **b** Gene-expression profiles of photosynthetic genes in spruce. **c** Expression specificity of biological processes. Rows indicate MapMan bins while columns indicate the species. The color intensity of the cells reflects the specificity of process expression, i.e., the number of time points where a bin is expressed above the average 80% of the maximum expression of the genes. **d** Percentages of rhythmic genes per biological process per species. The amounts are adjusted to ranges (≤20%, between 20 and 40%, between 40 and 60%, between 60 and 80% and >80%). The bars above both heatmaps summarize the processes expression

*TOC1* (Fig. 5b), which are typically expressed at a specific time point in the other species, are expressed more broadly in spruce. Whereas RNA sequencing data, due to providing only relative gene-expression abundances, cannot reveal whether the clock is inactive, the lower specific expression pattern of these key genes might explain the low frequency of rhythmic transcripts in the

spruce diurnal transcriptome. The basis of the weak diurnal rhythms observed in spruce (63°N[49]) is unclear, as another gymnosperm, Douglas-fir (42–43°N), shows substantially higher diurnal rhythms (29%[50]). Conversely, Japanese cedar (31–41°N) shows comparably weak diurnal rhythms (7.7%[29]). These results indicate that the weak diurnal rhythms in spruce are not general

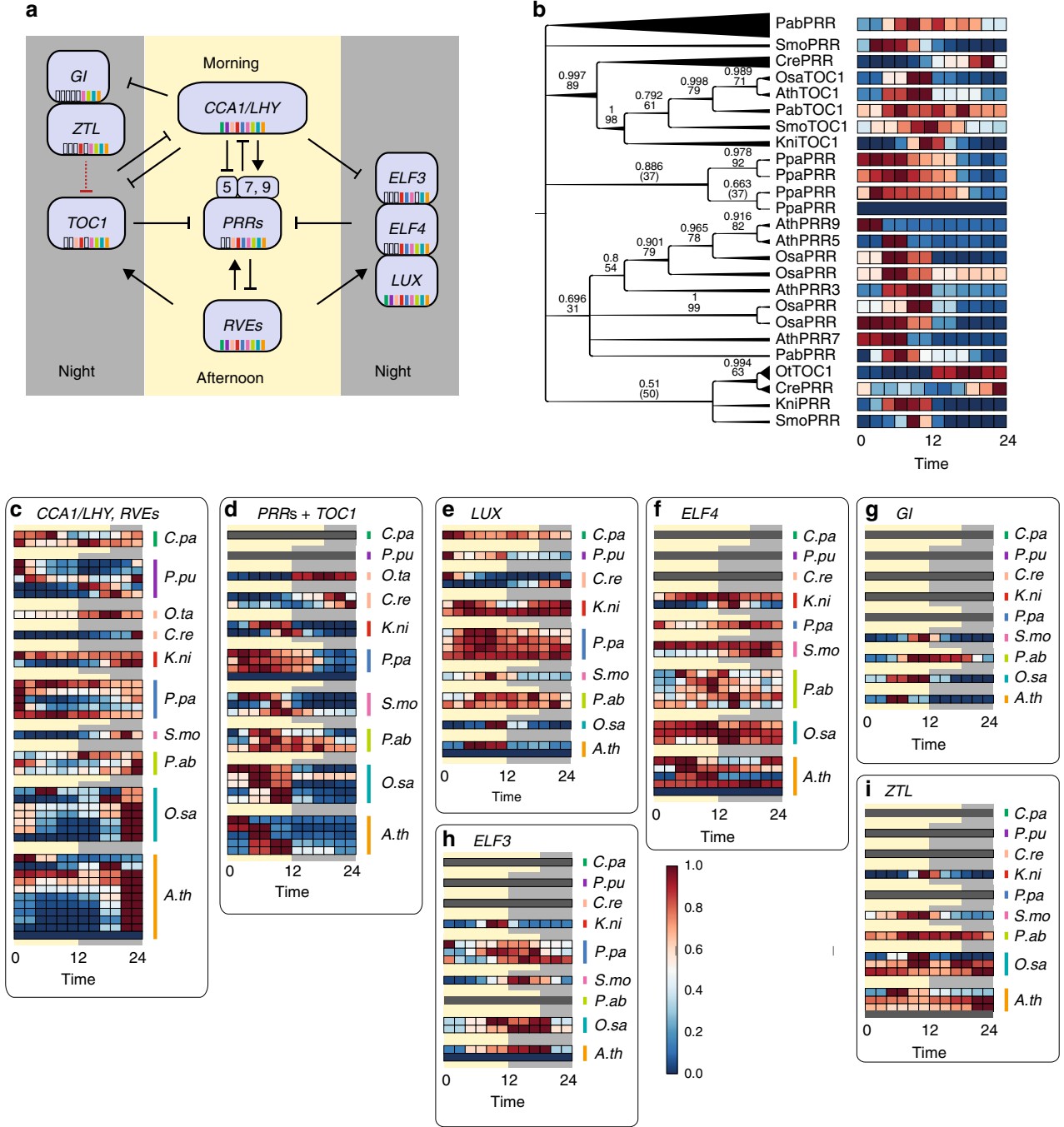

**Fig. 5** Analysis of clock core component genes. **a** Simplified *A. thaliana* circadian clock model (modified from ref. [39]). The colored bars within the gene families indicate in which plant clades a given gene family is present. The red dashed arrow indicate regulation at protein level. **b** Phylogenetic tree constructed on Bayesian inference of phylogeny. The numbers at the nodes indicate posterior probabilities (above) and bootstrap values (below). The gene annotation was based on[69]. The single heatmaps represent the 0–1-scaled expression profiles of each gene of the *PRR* family. Scaled expression profiles of genes belonging to **c** *CCA1/LHY/RVEs*, **d** *PRRs*, **e** *LUX*, **f** *ELF4*, **g** *GI*, **h** *ELF3*, and **i** *ZTL*. The scaled expression profiles are showing the expression for each time point collected

for the gymnosperm lineage, are not linked to the northern latitudes, but are rather a characteristic specific to spruce.

## Discussion

The life of photosensing organisms is finely tuned by photoperiodism, temperature, availability of photosynthetic products, and the circadian clock[51]. Our findings show that most members of the Archaeplastida subject more than one-third of their genes

to diurnal gene expression (Fig. 1b), underlining the importance of diurnal regulation. We demonstrated that, whereas the age of a gene family positively correlates with the level of rhythmicity and expression of genes (Fig. 2c, d), it does not affect diurnal expression patterns (Fig. 2a, Supplementary Fig. 2). This suggests that gene-expression levels and expression patterns are controlled by two separate mechanisms.

We found that diurnal programs are remarkably similar, despite the large phylogenetic distance spanning more than 1.5

billion years of evolution, and wide morphological diversity of the analysed species (Fig. 3e). Interestingly, diurnally regulated orthologs tend to be expressed at approximately the same time in the 24 h light–dark cycle, although the precise expression peak can shift offset by a few hours (Fig. 3e). This leads us to conclude that the sequence of the diurnal transcriptomes is similar because such an arrangement presents an optimal sequence of gene expression, regardless of morphology and habitat and was established early in Archaeplastida evolution. The exception to this rule is spruce which shows a largely arrhythmic expression. This suggests that despite its universality and ancient origin, diurnal gene expression can be suppressed to better suit a particular environment. However, we cannot exclude that in combination with the diurnal temperature oscillations, the amount of rhythmic genes would increase in spruce.

We observed that unicellular organisms exhibit a more specific diurnal gene expression, often restricting the expression of genes related to the major biological pathways to an 8-h period (Fig. 4c), and with a higher rhythmicity of basal processes, such as photosynthesis, major carbohydrate metabolism, oxidative pentose pathway, and tetrapyrrole synthesis. Conversely, *K. nitens* and land plants exhibit a more uniform expression of biological processes. We hypothesize that the increased complexity of multicellular organisms necessitated an uncoupling of the diurnal regulation of biological processes from the external influence of the light/dark alternation. While DNA replication in young seedlings is regulated by the clock[33], multicellular organisms in our study showed a more uniform expression of cell division markers (Fig. 1d), suggesting that cell division in mature tissues differs from young tissues. Furthermore, we cannot exclude that the combination of signals from the diurnal transcriptomes of the different cell types obscures the specific diurnal expression profiles of, e.g., epidermis and mesophyll cells[52,53]. Indeed, it has been observed that actively dividing cells in the shoot apex show more robust rhythmicity than roots, indicating that different tissues can be under alternative diurnal regulation[54]. However, the *K. nitens* data provide strong evidence for light-controlled cell division (Fig. 1d) in this species that possesses only a single cell type, but similar to land plants, shows a more uniform expression of biological processes. This result suggests that the more uniform expression pattern evolved in streptophytes. However, additional diurnal expression data from other Charophyceae, such as the recently sequenced *Chara braunii*[55], is needed to support this observation.

Our analysis confirms the ancient origin of the circadian clock by identifying *CCA1*- and *LUX*-like genes in glaucophytes and rhodophytes (Fig. 5c, e, Supplementary Figs. 8 and 10). However, the other essential components of the three main control loops are absent in these two species, suggesting that their clock might function differently from the well-studied clock in streptophytes. Additional studies, with experimental setups that minimize the influence of cell division and photoperiodism are needed to understand how the clock functions in these anciently diverged algae.

To further explore if the conservation of diurnal regulation is found across different tissue types, developmental stages, as well as protein and metabolite levels will be an interesting and promising area of future investigation.

## Methods
**Growth conditions**. *Cyanophora paradoxa* UTEX555 (SAG 29.80, CCMP329) was obtained from Provasoli-Guillard National Center for Marine Algae and Microbiota (NCMA, Bigelow Laboratory for Ocean Sciences). The cultures were grown in C medium[56], at 24 °C in 16L/8D photoperiod (40 μmol photons m$^{-1}$ s$^{-1}$), aerated with normal air. *P. purpureum* SAG 1380-1d was obtained from SAG (Culture Collection of Algae at Göttingen University). The cultures were grown in ASW medium[57], at 25 °C in 12L/12D photoperiod (100 μmol photons m$^{-2}$ s$^{-1}$). *K.*

*nitens* (NIES-2285) was obtained from the NIES collection (Tsukuba, Japan). The cultures were grown in liquid C medium at 24 °C, in 12L/12D photoperiod (40 μmol photons m$^{-2}$ s$^{-1}$). *P. patens* (Gransden wild type) was grown in Jiffy −7 peat pellets at 23 °C, in 16L/8D photoperiod (100 μmol photons m$^{-2}$ s$^{-1}$) and the gametophytes were collected. *S. moellendorffii* (clone of the sequenced *S. moellendorffii*) was grown on modified Hoagland solution (0.063 mM FeEDTA; 0.5 mM KH$_2$PO$_4$; 2.5 mM KNO$_3$; 2 mM Ca(NO$_3$)$_2$ × 4H$_2$O; 1 mM MgSO$_4$ × 7H$_2$O; 50.14 μM H$_3$BO$_3$; 9.25 μM MnCl$_2$ × 4H$_2$O; 1 μM ZnCl$_2$; 1 μM CuCl$_2$; 0.5 μM Na$_2$MoO$_4$ × 2H$_2$O, pH 5.8, 0.8% agar) at 24 °C, in 12L/12D photoperiod (70 μmol photons m$^{-2}$ s$^{-1}$) and microphylls were collected. All organisms were actively growing. *P. abies* seedlings (offspring of the Z4006 clone, sampled from Ragunda, central Sweden) were grown in soil at 24 °C, in 16L/8D photoperiod (250 μmol photons m$^{-2}$ s$^{-1}$) and needles were collected.

**RNA isolation and RNA sequencing**. The total RNA was extracted using Spectrum™ Plant Total RNA Kit (Sigma-Aldrich) according to the manufacturer's instructions. The integrity and concentration of RNA was measured using RNA nano chip on Agilent Bioanalyzer 2100. The libraries were prepared from total RNA using polyA enrichment and sequenced using Illumina-HiSeq2500/4000 at Beijing Genomics Institute and Max Planck-Genome-centre in Cologne.

**Analysis of RNAseq data and microarrays data**. The reads were trimmed, mapped, counted and TPM normalized using the LSTrAP pipeline[58]. The genomes used for mapping are for *C. paradoxa* v1.0 (unpublished genome update can be obtained from Debashish Bhattacharya, d.bhattacharya@rutgers.edu, U Rutgers), *P. purpureum* v1.0 (obtained from *The Porphyridium purpureum Genome Project*)[59], *K. nitens* v1.0 (obtained from *K. nitens* NIES-2285 genome project)[60], *P. abies* v1.0 (obtained from PlantGenIE)[49], and *P. patens* v1.6 (obtained from Cosmoss)[4]. Reads from *S. moellendorffii* were mapped to the transcriptome. *C. reinhardtii* RNA-seq counts were obtained from ref. [17] and were TPM normalized. Only samples taken every two hours were considered for the analysis to better match the sample density of the other species. *Synechocystis* sp. PCC 6803 microarray raw data, obtained from ArrayExpress accession E-GEOD-47482[23] was processed using the R-package limma[61], while *A. thaliana* raw data, obtained from E-GEOD-3416[45] were RMA normalized. Processed *O. sativa* expression data was obtained from E-GEOD-28124[24] (seedling dataset). The quality of the samples and agreement of the replicates was assessed through PCA analysis. For further analyses only expressed genes were selected from the normalized matrices. A gene was selected as expressed if the TPM value was >1 for the RNAseq data. For *A. thaliana*, MAS5 software[62] had to call a gene "present" in at least two out of three replicates, while for *O. sativa*, in at least one out of two replicates.

For the comparison of rhythmicity in different developmental stages in *O. sativa* processed expression data was obtained from E-GEOD-28124[24] (flag leaf dataset) and *A. thaliana* raw data were obtained from E-MEXP-1304[63] and RMA normalized.

**Detection of rhythmic genes**. Rhythmic genes were identified by using the JTK algorithm[64]. For *C. paradoxa*, *P. purpureum*, *C. reinhardtii*, *K. nitens*, *P. patens*, *S. moellendorffii*, and *P. abies* we used a total of 12 time points (ZT1 to ZT23), and the replicate number was set to 3 (2 for *C. reinhardtii*). The temporal spacing between samples was set to 2 h and the target period was set to 12 h (effectively equal to 24 h). The *Synechocystis* sp. PCC 6803 dataset consisted of one replicate per time point over 2 day cycle for a total of 24 time points, and the JTK parameters were set to: number of replicates to 1 and the target period 48 h (effectively equal to 24 h). For *A. thaliana* and *O. sativa* the temporal spacing between samples was set to 4 h and the target period to 6 h (effectively equal to 24 h). For *A. thaliana*, the number of replicates was 3 while for *O. sativa* 2. Using an adjusted $p$ value cutoff < 0.05 we identified 2440, 12343, 6597, 12341, 9375, 11692, 11860, 5280, 8082, and 5676 rhythmic genes for *Synechocystis* sp. PCC 6803, *C. paradoxa*, *P. purpureum*, *C. reinhardtii*, *K. nitens*, *P. patens*, *S. moellendorffii*, *P. abies*, *O. sativa*, and *A. thaliana*, respectively (Supplementary Datas 2–11). JTK algorithm shows a different sensitivity in rhythmic genes detection which correlates with the sample density and number of replicates[65], therefore, in order to calculate the percentages of rhythmic genes in a comparable manner we performed an additional run of JTK where the initial expression matrices were adapted to the dataset with the least number of time points and replicates (*O. sativa*). For this purpose we set the number of replicates to 2, the spacing between samples to 4 h and the target period was set to 6 h (Supplementary Fig. 14). We observed an average decrease of 26% of rhythmic genes when sampling density was decrease from 12 daily time points to 6 daily time points, indicating that the amount of rhythmic genes is most likely higher than we report. LAG 0 was identified as the first time point collected after the light was turned on (i.e., LAG 0 in *C. paradoxa* was corrected to LAG 1, LAG 0 in *A. thaliana* was corrected to LAG 4).

To validate the results obtained by JTK we used the Haystack algorithm[66] (Supplementary Fig. 15). The correlation cutoff was set to 0.7, the fold-change to 1, the background cutoff to 1 and a $p$ value cutoff of 0.05. The used five models to infer the rhythmicity. We detected 3004, 3756, 6354, 13429, 5630, 3059, 12914, 20613, and 12507 rhythmic genes for *Synechocystis* sp. PCC 6803, *C. paradoxa*, *P. purpureum*, *C. reinhardtii*, *K. nitens*, *P. patens*, *S. moellendorffii*, *P. abies*, *O. sativa*,

and *A. thaliana*, respectively. The analysis showed a good agreement between the significantly rhythmic genes, and their estimated phases (Supplementary Fig. 15).

**Detection of orthologous genes with Orthofinder**. Orthologous genes were obtained by using OrthoFinder v1.1.8[35] with default parameters and Diamond[67], which identified 12,913 orthogroups (Supplementary Data 14). For further analysis, we considered only one-to-one orthologs, in order to ensure that the comparison between genes would dismiss comparisons between unclear orthologs.

**Phylostratigraphic analysis of rhythmicity and expression**. The phylostratum of each orthogroup was estimated by finding the oldest clade in the family. To estimate if a given phylostratum is significantly rhythmic, we first counted how many genes are rhythmic in a phylostratum. Next, we performed a permutation analysis where we shuffled the gene-phylostratum assignment 1000 times, and we obtained a number of rhythmic genes per phylostratum. The empirical *p* value for enrichment was obtained by counting the number of times the permuted distribution showed a higher number of rhythmic genes compared to the observed values, divided by the number of permutations (1000). The empirical *p* values for depletion were calculated in the same manner.

The gene average expression of the phylostrata was obtained from the average gene expression of each gene assigned to each phylostratum. To identify average expressions that are significantly different, we used a two-sample K–S test for all possible combinations. The obtained *p* values were FDR corrected. The results of the comparison are found in Supplementary Fig. 3.

**Estimating the similarity of diurnal transcriptomes**. To estimate the similarity of diurnal transcriptomes of two species, we first calculated the absolute difference of expression ($\Delta$phase$_{observed}$) among each pair of one-to-one orthologs. Because of the cyclic nature of the data, for $abs(\Delta$phase$_{observed}) > 12$, we used $abs(\Delta$phase$_{observed} - 24)$. Next, we calculated this number for all one-to-one orthologs to arrive at the average $\Delta$phase$_{observed}$. Finally, to test whether a similar distribution of pairs would have been obtained by chance, we shuffled the columns (time points) in the expression matrices 1000 times, ran JTK for each permutation, and obtained a phase value for each gene. For each permutation, we calculated the $\Delta$phase value, as described above, to arrive at 1000 $\Delta$phase$_{expected}$ values. The empirical *p* value was calculated by counting how many times the $\Delta$phase$_{observed}$ was smaller than the $\Delta$phase$_{expected}$ values. The *p* values were then FDR corrected.

The shift analysis was performed by adding an integer in the interval of $[-12, 12]$ to the phase values of all the rhythmic genes of the older species, and calculating $\Delta$phase$_{observed}$ upon the shift (here defined as $\Delta$phase$_{shifted}$). For each shift, the empirical *p*-value was calculated, as described above, and the $\log_2(\Delta$phase$_{expected}/\Delta$phase$_{shifted})$ was calculated. The first quartile, the median and the third quartile of the distribution was plotted and color coded according to indicate significantly (empirical *p* value < 0.05, orange, red) and not significantly (gray, black) similar transcriptome upon the shift. The shift which provided the highest $\log_2(\Delta$phase$_{expected}/\Delta$phase$_{shifted})$ was used to identify the lowest *p* value (Fig. 3c). The *p* values corresponding to the best shift for each species comparison were then FDR corrected (Fig. 3e).

**Diurnal gene expression of biological processes**. We used Mercator with standard settings to annotate the coding sequence with MapMan bins[68]. To investigate the expression of genes involved in the main biological processes, we analyzed the more general first level MapMan bins. For each gene in a bin, the expression vector of the gene was first scaled by dividing the vector by the maximum value of the vector. Then, the expression vectors of the genes within the bin were summed, and the summed vector was divided by the maximum value. Finally, to calculate the specificity of the MapMan bin expression, we counted the number of time points with value >0.8.

**Rhythmicity of biological processes**. The rhythmicity of the processes was studied by calculating the percentage of rhythmic genes assigned to a given first level MapMan bin. To make the results comparable between the 4 h (*A. thaliana* and *O. sativa*) and 2 h experiments (every other Archaeplastidia member), we removed every other time point from the 2 h expression data, to arrive at 4 h expression matrices. The expression matrices were then analysed by JTK to arrive at rhythmic genes, which were used to generate Fig. 4d.

**Phylogenetic analysis and tree construction**. To add *C. paradoxa* and *P. purpureum* to the clock gene families, we identified eight orthogroups (OG0000138, OG0000233, OG0001151, OG0001171, OG0001580, OG0003526, OG0007425, and OG0008362) that contained the 20 major clock components from *A. thaliana* and we inspected protein alignments of these two gene families with the *A. thaliana* proteins, to identify the clock orthologs in the two algae[69]. The protein sequences were aligned using MUSCLE software[70] (Supplementary Data 15, alignments for the Figure obtained from M-COFFEE[71]) and trimmed with TrimAl with standard settings[72]. The best-fit models for each orthogroups were identified using ProtTest 3.4.2[73], which was based on the Bayesian information criterion. The phylogenetic reconstruction was done by analyzing the protein alignments with both Maximum

Likelihood (ML) and Bayesian inference (BI) methods. The ML analysis was performed using PhyML 3.0 algorithm[74] with a bootstrap with 100 iterations. Bayesian analysis was conducted using MrBayes 3.2.6[75] where two Markov Chain Monte Carlo (MCMC) runs were performed for 1 million generations, and sampling was performed every 500 generations. Both the average standard deviation of split frequencies and potential scale reduction factor were <0.01 and close to 1, respectively, across the two runs. The trees were visualized using TreeGraph2[76] using the Bayesian tree as a reference. The bootstrap values from PhyML and the posterior probabilities values from MrBayes were mapped as branch support. For large orthogroups, such as *PRRs* (OG0000138), *CCA1/LHY-RVEs* (OG0000233), and *ZTL* (OG0001151), we isolated the relevant clades by analyzing phylogenetic trees based on ML and BI methods (Supplementary Figs. 8–13), and the phylogenetic analysis was run on the selected genes. As previously reported[69], we could not separate *CCA1/LHY* from *RVE* genes since both classes of proteins are MYB-like transcription factors[77–79] (Supplementary Fig. 8). Due to the extensive characterization of *O. tauri* circadian clock, we included the two genes representing *CCA1* and *TOC1* in the phylogenetic tree analysis. Expression data were obtained from E-GEOD-16422[19].

## Data availability

The TPM normalized expression data are available in Supplementary Datas 2–11, while the fastq files representing the raw RNA sequencing data have been deposited in EBI Array Express under accession number E-MTAB-7188. *Synechocystis* sp. PCC 6803, *O. tauri*, *O. sativa*, *A. thaliana* (seedlings and rosette) data are available from EBI Array Express under accession number E-GEOD-47482, E-GEOD-16422, E-GEOD-28124, E-MEXP-1304, E-GEOD-3416, respectively. *C. reinhardtii* data is available at the NCBI Gene Expression Omnibus repository under accession number GSE71469.

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

## Acknowledgments

We would like to thank the employees of the Max Planck Institute of Molecular Plant Physiology and Nanyang Technological University for support. The sequencing resources and other funding were provided by the Max Planck Gesellschaft and ERA-CAPS project EVOREPRO (S.P.). We thank Dr. Gopikrishna J. (gopikrrishna.j@gmail.com) for the illustration in Fig. 1a.

## Author contributions

M.M. conceived the project; C.F., M.M. and S.P. performed the bioinformatical analyses with inputs from Z.N. and M.S.; M.J. provided *Selaginella moellendorffii* cultures; J.B. provided *Physcomitrella patens* cultures; D.B. and D.P. provided the updated *Cyanophora paradoxa* genome; T.T., A.B.E., M.S. and A.F. provided feedback on the manuscript and experimental design. C.F. and M.M. wrote the article with the help from all the authors.

## Additional information

**Competing interests:** The authors declare no competing interests.

