## [Peer Review File · Nature Communications]

Reviewers' comments:

Reviewer #1 (Remarks to the Author):

It is broadly appreciated that the diurnal light-dark cycle has exerted a profound influence on the behavior of virtually all life and that circadian clocks have evolved to enable organisms both to synchronize with the environmental daily and seasonal cycles and to anticipate the major daily transitions of dawn and dusk. Although most work on plant clocks has emphasized angiosperms, and especially the model plant *Arabidopsis thaliana*, circadian rhythms have been demonstrated in green algae such as *Chlamydomonas reinhardtii* and in the picoalga *Ostreococcus tauri*. However, prior to this work, there has been no systematic survey of the extent of diurnal control of the transcriptome across the plant lineage. They compare transcriptomes of the cyanobacterium, *Synechocystis*, and 9 members of the Archaeplastida, each representing a key node in the lineage. They used publicly available data for 4 transcriptomes and generated 6 new transcriptome data sets. Their analysis has several major results:

First, the diurnal transcriptomes are surprisingly similar across the spans of evolutionary distance, morphology, and habitat. Roughly 1/3 of the transcriptome cycles in each taxon except spruce, where that number is much lower, consistent with earlier analyses of photosynthetic gene expression. Second, the age of a gene family is positively correlated with both the level of rhythmicity and expression level, but does not influence the diurnal expression pattern. This allows the authors to conclude that expression level and expression pattern are subject to distinct control mechanisms. This, I think, is an important insight.

Third, their analysis shows that the loss of clock-controlled cell division was associated with the origin of multicellularity, rather than with the emergence of land plants.

I think this is an extremely important piece of work. I think it makes some very important points about the evolution of temporal control of gene expression and I think it will be of broad interest and will be highly-cited. I enthusiastically support publication. I have no major criticisms.

Minor points:

1. The description of the analysis of phase I found to be somewhat confusing. They refer to shifting lags forward and so on. I think they need to be more precise in their language and perhaps provide a concrete example. Cyanophora genes tend to peak 2 h earlier than *Chlamydomonas*. So Cyanophora genes do not lag. So shifting the lags of Cyanophora gene forward by two hours confuses me. To me, forward means earlier, rather than later. For example, the clock field would say that the Cyanophora phases are advanced relative to the *Chlamydomonas* phases. Moving those Cyanophora phases forward by 2 hours to me would make them not 4 h earlier than the *Chlamydomonas* phases. So I think this description needs to be rewritten to be unambiguous.
2. It has been suggested that LHY is the ancestral gene and CCA1 is derived from a duplication event (Lou et al 2012 Plant Cell). Does their phylogenetic analysis (SF6) permit this confirmation or rejection of this hypothesis and does it speak to the relative ages of LHY/CCA1 versus the RVEs?
3. Why in their phylogenetic analyses is TOC1 considered separately from the other PRRs?

Reviewer #2 (Remarks to the Author):

Review to "Kingdom-wide comparison reveals the evolution of diurnal gene expression in Archaeplastida" by Ferrari et al., submitted to Nature Communications.

Ferrari et al. have used a wide systemic approach to unravel the evolution of the circadian clock from primitive algae to modern plants, using a cyanobacteria as an outgroup control. The conclusions, although most of them have already been reached by other researchers, are interesting because a much wider approach has been used than before. Still, some of the conclusion needs to be reviewed, as well as some of the statements throughout the text.

Major points.

1. The cell cycle is synchronized with the end of the day in algae in synchronous cultures, and clear markers for this are the peak in expression of cyclins and cyclin-dependent kinases. This kind of division has not been conserved in plants, as has been described before. These would be much better than DNA replication markers. Are CYCs and CDKs genes enriched in your algal analysis? In Fig 1D, what is the presence of these genes, are they circadian in plants and not cycling in plants?
2. There is a recurrent conclusion that the diurnal regulation of gene expression in spruce is lower compared with the rest of species. A generalization to all gymnosperms is based on this conclusion throughout the text. *Picea albies* has a broad range of latitude and altitude habitats, and it has been shown that its clock is very dependent not only on light but on temperature (as cited for *Arabidopsis*). Nevertheless, in experimental procedures it is said that *Picea* is grown at 24 C continuously, without day/night changes which are very common and often extreme in their habitats. Therefore, it is very possible that the low dependence of the clock on light is because the compensation by temperature is lost. Without a temperature compensation the authors can only stress that the light input is low to the clock but not that the clock regulation of gene expression is low. This must be altered throughout the text.
3. Discussion in lines 238-253 are very speculative. When comparing the circadian expression of two different species grown on different light regimes, the differences referred by the authors may reflect exactly these differences in day-light regimes, and not a particular characteristic of the species' circadian clock. Without a control in which both species are grown and sampled in the same experimental frame of light, photoperiod and temperature, only predictions and assumptions, can be made.
4. While I tend to agree with authors that Charophyceae developed a special gene expression mechanisms adapted to multicellularity, the species used in this experimental approach, is only filamentous and single-celled. Without circadian data from a real "talophytic" algae, such as the recently sequence "*Chara braunii*" (Nishiyama et al, 2108, Cell vol 174) this would be very speculative. I would rather "suggest" and insist in this lack of data.
5. In microalgae, ADAGIO-LIKE genes such as ZTL or FKF1 do not show a complete domain structure, rather the Fbox, Kelch repeat and LOV domains are separated in two or three proteins. It is curious that a rather ancient clade such as the Glaucophytes would have so clear a ZTL orthologous. Authors should discuss on this remarkable discovery or check the data and see if the protein structure is like a true ZTL protein.

Minor points

- References for synchronous cultures in algae could be more up to date than the 1979 ref.
- The concept of "old" genes should be better explained by authors, as genes themselves have no age, are rather the result of longer or shorter evolutionary processes in any particular species and is difficult to generalize.
- In several figures (such as in 2B) please define the X-axis scale and show the scale independently in every figure, not just in supplemental files.
- It would be informative to indicate in Figure 3E which percentage of the analysis defers from the diagonal distribution, which is the statistical approach of this significance and which percentage of genes are diagonally distributed in each species. A look at the difference figures shows that actually most of the genes are sited out of this diagonal distribution.
- Line 826 please correct "indicates".
- Except when deliberately made a common name (and then define first time mentioned), please use genera and species names in italics.

- Line 286 and thereon. In this discussion the word “ubiquitous” is not well understood. Are authors referring to space ubiquity or to ubiquitous expression in time. Or rather, it is being used in a rather subjective way referred to the way the data in the figure is organized?
- Supplemental Fig 6. Should not the colored panels on the right have a 24h time scale?
- Regarding one of the last conclusions, the identification of CCA1/LHY orthologous proteins in Cyanophora and Porphyridium is quite significant. Have you tried to use these proteins and search in cyanobacteria databases to see if the genes precede in some cyanobacteria divisions the symbiotic event? This would be a very interesting point in the clock evolution.
- Fig 5. Please explain and show x-scale on the right panels.

Reviewer #3 (Remarks to the Author):

In this manuscript the authors present a comparative analysis of the dynamics of gene expression under light/dark cycles in different species from different Archaeplastida lineages and cyanobacteria. They use publicly available transcriptome data from a cyanobacterium (*Synechocystis* sp. PCC6803), a chlorophyte (*Chlamydomonas reinhardtii*) and land plants, (*Arabidopsis thaliana*, *Oryza sativa*) and generate transcriptome data for a glaucophyte (*Cyanophora paradoxa*), a rhodophyte (*Porphyridium pupureum*), a charophyte (*Klebsormidium nitens*) and a gymnosperm (*Picea abies*). They observe significant conservation of the phases of gene expression among orthologs, complementing single species studies in algae (Noordally and Millar 2015) and land plants (Filichkin et al., 2011, PMID: PMC3111414). This study also demonstrates that older genes are more likely to cycle than newer lineage specific genes. The authors also provide analysis of the orthologs of land plant clock genes. The authors conclude that there is a loss of diel control of cell cycle in multicellular species and general decreased diel control of gene expression in multicellular organisms.

The authors generate an interesting and valuable dataset for comparative analyses of diel expression in photosynthetic organisms. My main concern is their conclusion on loss diel regulation of the cell cycle and overall gene expression in multicellular organisms.

They observe that the percentage of cell cycling genes that had diel oscillations was lower in the multicellular organisms. The tissue sampled from land plants probably contained very few dividing cells since they were, I believe, from mature leaves (although the plants were actively growing). This is likely very different from unicellular algae, in which, depending on the light intensity and nutrient status a significant percentage of cells will be dividing at a given cycle. It has been recently shown that in young *Arabidopsis* seedlings DNA replication is regulated by the circadian clock (Fung-Uceda et al., PMID:29576425). In addition, meristems in *Arabidopsis* appear to have more robust rhythms than other tissues (Takahashi et al., PMID:26406375). Therefore, for a proper comparison actively dividing cells from land plants would need to be compared to actively dividing cells from unicellular algae. The apparent lower rhythmicity of *Klebsormidium*, speaks for a role of multicellularity per se, since this charophyte lacks differentiated cells. However, this observation is on only one species measured under one condition, which is not sufficient to prove such a strong and important conclusion. In the absence of experimental studies on the synchronization of cell division in the species studied, the authors should provide a more nuanced interpretation of this result in the abstract and at the end of the introduction, clarifying the different alternative hypothesis.

The authors also observe that the phases of gene expression in multicellular organisms are more widely distributed than in single celled algae. They conclude that this result indicates a less “defined diurnal control”. However, it could also be interpreted as a more sophisticated fine-tuned diurnal control in which time can be measured and defined with more precision. This might be necessary to

adjust the needs of different cell types. Although the circadian regulatory network in green algae is not well characterized the increase in the number of circadian clock components in land plants (as the authors show) suggests a need of a more complex regulation of time dependent expression. It is also important to take into consideration that the phase distribution of gene expression varies under different growth conditions (Michael et al., PMID: PMC2222925) and based on the developmental stage of the organ, for example older leaves in rice have a wider more monophasic distribution of phases than young seedling leaves (Xu et al., 2011). It would be interesting to see how diel transcriptomic datasets from Arabidopsis seedlings (such the ones from Michael et al., 2008) differ from mature leaves. Here again a more nuanced interpretation of the results is necessary and additional analyses might help clarify this point.

In general, the results require a deeper and more nuanced interpretation and discussion. Below are some suggestions.

Other comments:

1. It would be interesting to know which are the genes (orthologs) with highly conserved expression patterns. Are there any processes with more of those genes? That will help understand which processes require cyclic gene expression.

2. Line 140: The authors use "lag" values to describe the phase of expression. Phase is a more commonly used term. Phase should be represented as zeitgeber time (ZT) which is "hours after dawn" a metric commonly used in chronobiology. The "lag" values appear at first to be defined as time point of the series, ie. it initially appears that lag 3 would not be the same in a 2 h resolution experiment than at 4 h resolution experiment. The use of "phase" and ZT would be less ambiguous.

2. Line 151/Line 380: The authors observe that *P. abies* have significant lower levels of cyclic gene expression. The authors hypothesize that this is due to its origin at very high latitudes and therefore exposure to extreme changes in photoperiod. The interpretation/discussion of this result needs to be improved and the following points addressed:

a. This argument is often used on the opposite way, such that exposure to changes in photoperiod require the presence of a strong circadian oscillator. Is there any evidence for weaker diel rhythms in species originating from northern latitudes?

b. There are many *P. abies* genotypes and some are surely adapted to more southern regions. It would be important to know the ecotype used.

d. Gyllenstrand et al., 2014 (PMID:24363286) showed that in *P. abies* (L. Karst) the amplitude of diel expression of some circadian related genes is significantly higher under short days than under long days. Maybe photoperiod has a general influence of rhythmicity in spruce.

On Line 353 the authors write "suggesting that the clock is running differently or is not active in spruce". What do you mean "differently"? . As shown by Gyllenstrand it is possible that a different photoperiod or other condition (temperature, light intensity) leads to stronger rhythms.

c. There are other studies on diel transcriptome expression in gymnosperms. Nose & Watanabe (2014) (PMID: 25403374) analyzed the diel expression in Japanese cedar and concluded that about ~7% of the genes cycled. Cronn et al., (2017) analyzed diel expression in Douglas-fir needles and showed that ~30% of genes cycle. That is similar to another tree, poplar (Filichkin et al., 2011). These other experiments should be

3. Line 161: what do you mean with double peaks.

4. Line 193: The authors observe that there is a significant tendency for genes from older phylostrata to have higher rhythmicity than genes from younger phylostrata.

This agrees with a study using *Chlamydomonas* that demonstrated that there is a significant tendency of cyclic gene expression to be maintained after gene duplication (Panchy et al., PMID: 25354782).

5. Line 207 " We therefore conclude that older phylostrata tend to be more strongly expressed". Is that because these genes are expressed in more tissues?

6. Line 232: The DeltaLag description is confusing because it is unclear when a distribution or the average is described. Does this sentence (on line 232) mean:
"The analysis revealed that, in contrast to evenly distributed DeltaLagExpected values with an average DeltaLagExpected of 5.98 the distribution of DeltaLagObserved values peaked at +2 hours and the average DeltaLag Observed was 4.07."?

7. The *Chlamydomonas* original experiment had 1 h resolution sampling, but on line 447 the authors indicate that they use samples taken every 2 h. I assume that this is to match better the other studies, but an explanation is needed. Related to sampling frequency, the authors should mention that an increase sampling density increases the number of genes classified as cycling using JTK cycle (Li et al., 2015; PMID:25662464) and they should discuss whether the differences in their experiments could influence their results about the percentage of cycling genes.

8. Line 272: "with an anticipatory decrease below 50% of maximum expression before dusk", Since the expression of FBPsases and RPE increase just before dusk, I would change this description to "with a decrease below 50% of maximum expression during the second half of the light period".

9. Line 316: It has been demonstrated that in *Arabidopsis* CCA1 and LHY repress the expression of the PRRs (Kamioka et al., 2016; PMID: 26941090). That needs to be corrected on Figure 5A as well.

10. Line 322: It is misleading to say that GI and ZTL repress the expression of TOC1 and other PRRs genes. GI and ZTL are involved in the degradation of TOC1 and PRR5 proteins, not in the transcription of the respective genes.

11. Line 333: I would not call it an "incomplete" Viridiplantae clock, I would say "despite of the lack of conservation of the Viridiplantae clock". Since it is possible that these algae have a circadian clock as the authors indicate at the end of that sentence.

12. Figure 5B1: TOC1 proteins should be included in the PRR tree. TOC1s are PRR proteins and in the green algae "TOC1s" cannot be assigned into either "TOC1" or "other PRRs" (Linde et al., 2017; Liu et al., 2013 PMID:23856081). The description of the PRRs in the text should include TOC1 and other PRRs, trying to define a *Chlamydomonas* TOC1 is misleading.

13. Figure 5H: why are there two TOC1 genes in *Arabidopsis*?

14. Sup. Fig. 7 and 8 trees should be combined in one.

15. It would have been good to include *Ostreococcus tauri* data because in this green alga the function of "conserved" clock components CCA1/LHY/RVE and PRR have been described. Although clock components have been identified in *Chlamydomonas* the clock network is poorly characterized.

16. Line 358: "the lower specific expression pattern", do you mean the broader peaks of expression? Sequencing data not only provides relative expression abundances, it allows to determine the overall expression level of genes relative to others, is the expression of clock genes in spruce lower than in other plants?

17. Methods.

- a. Which data set from rice was used? It seems that in the study from Xu et al., 2011 to types of leaf, flag leaf and seedling leaf were used. The both types of leaves appear to have slightly different phases of gene expression, with the seedling leaves displaying a more narrow range of phases.
- b. The strain/ecotype information of all the species used needs to be included.
- c. Line 456: why was there a different in "detection" threshold for Arabidopsis and Oryza, please include an explanation.
- d. The new transcriptome data should be deposited in a public database.

18. Minor edits:

Figure 1C and D, font too small

Figure 2 fonts too small

Figure 3 the smaller fonts are too small

Figure 5A: what do the colored lines in the gene symbols mean?

Figure 5 B tree, font too small

Line 659: in this reference, check name of author

References: several references lack page numbers

Reviewers' comments:

Reviewer #1 (Remarks to the Author):

It is broadly appreciated that the diurnal light-dark cycle has exerted a profound influence on the behavior of virtually all life and that circadian clocks have evolved to enable organisms both to synchronize with the environmental daily and seasonal cycles and to anticipate the major daily transitions of dawn and dusk. Although most work on plant clocks has emphasized angiosperms, and especially the model plant *Arabidopsis thaliana*, circadian rhythms have been demonstrated in green algae such as *Chlamydomonas reinhardtii* and in the picoalga *Ostreococcus tauri*. However, prior to this work, there has been no systematic survey of the extent of diurnal control of the transcriptome across the plant lineage. They compare transcriptomes of the cyanobacterium, *Synechocystis*, and 9 members of the Archaeplastida, each representing a key node in the lineage. They used publicly available data for 4 transcriptomes and generated 6 new transcriptome data sets. Their analysis has several

major results:

First, the diurnal transcriptomes are surprisingly similar across the spans of evolutionary distance, morphology, and habitat. Roughly 1/3 of the transcriptome cycles in each taxon except spruce, where that number is much lower, consistent with earlier analyses of photosynthetic gene expression.

Second, the age of a gene family is positively correlated with both the level of rhythmicity and expression level, but does not influence the diurnal expression pattern. This allows the authors to conclude that expression level and expression pattern are subject to distinct control mechanisms. This, I think, is an important insight.

Third, their analysis shows that the loss of clock-controlled cell division was associated with the origin of multicellularity, rather than with the emergence of land plants.

I think this is an extremely important piece of work. I think it makes some very important points about the evolution of temporal control of gene expression and I think it will be of broad interest and will be highly-cited. I enthusiastically support publication. I have no major criticisms.

Response: We thank the reviewer for the very enthusiastic appreciation of our work and for the valuable comments.

Minor points:

1. The description of the analysis of phase I found to be somewhat confusing. They refer to shifting lags forward and so on. I think they need to be more precise in their language and perhaps provide a concrete example. Cyanophora genes tend to peak 2 h earlier than *Chlamydomonas*. So Cyanophora genes do not lag. So shifting the lags of Cyanophora gene forward by two hours confuses me. To me, forward means earlier, rather than later. For example, the clock field would say that the Cyanophora phases are advanced relative to the *Chlamydomonas* phases. Moving those Cyanophora phases forward by 2 hours to me would make them not 4 h earlier than the *Chlamydomonas* phases. . So I think this description needs to be rewritten to be unambiguous.

Response: We agree with the reviewer on this point, and to make the text clearer, we have replaced 'lag' with 'phase'. To make this point more clear, we have added this sentence: 'To this end, we compared the observed phase differences (\$\Delta\text{phase}_{\text{observed}}\$ ) with the phase differences calculated from permuted expression data (\$\Delta\text{phase}_{\text{expected}}\$, Fig. 3B, light blue

bars). Indeed, the analysis revealed that, in contrast to evenly distributed $\Delta phase_{expected}$ values, the $\Delta phase_{observed}$ values peaked at 2-3 hours (Fig. 3B). This indicates that *C. paradoxa* reaches the highest similarity to *C. reinhardtii* when its phases are delayed by two hours.’

2. It has been suggested that LHY is the ancestral gene and CCA1 is derived from a duplication event (Lou et al 2012 Plant Cell). Does their phylogenetic analysis (SF6) permit this confirmation or rejection of this hypothesis and does it speak to the relative ages of LHY/CCA1 versus the RVEs?

Response: The reviewer raises an interesting point. However, the SF8 containing the CCA1 and LHY phylogenetic tree does not allow us to make any statements about the relative ages of these genes.

3. Why in their phylogenetic analyses is TOC1 considered separately from the other PRRs?

Response: Due to the points raised by Reviewers 2 and 3, we are not using OrthoFinder gene families for the clock analysis, as OrthoFinder gene families showed minor, but significant inconsistencies with literature. We are now using gene families based on literature, where TOC1 and PRRs are found in the same family (Figure 5 and SF9). We thank the reviewer for spotting this inconsistency.

Reviewer #2 (Remarks to the Author):

Review to “Kingdom-wide comparison reveals the evolution of diurnal gene expression in Archaeplastida” by Ferrari et al., submitted to Nature Communications.

Ferrari et al. have used a wide systemic approach to unravel the evolution of the circadian clock from primitive algae to modern plants, using a cyanobacteria as an outgroup control. The conclusions, although most of them have already been reached by other researchers, are interesting because a much wider approach has been used than before. Still, some of the conclusion needs to be reviewed, as well as some of the statements throughout the text.

Response: We thank the reviewer for the very useful and constructive comments.

Major points.

1. The cell cycle is synchronized with the end of the day in algae in synchronous cultures, and clear markers for this are the peak in expression of cyclins and cyclin-dependent kinases. This kind of division has not been conserved in plants, as has been described before. These would be much better than DNA replication markers. Are CYCs and CDKs genes enriched in your algal analysis? In Fig 1D, what is the presence of these genes, are they circadian in plants and not cycling in plants?

Response: This is a very good suggestion, and we have modified Figure 1D accordingly:

As seen on the figure, the cyclin and cyclin-dependent kinases show similar expression profiles to the DNA replication markers.

2. There is a recurrent conclusion that the diurnal regulation of gene expression in spruce is lower compared with the rest of species. A generalization to all gymnosperms is based on this conclusion throughout the text. *Picea alba* has a broad range of latitude and altitude habitats, and it has been shown that its clock is very dependent not only on light but on temperature (as cited for *Arabidopsis*). Nevertheless, in experimental procedures it is said that *Picea* is grown at 24 °C continuously, without day/night changes which are very common and often extreme in their habitats. Therefore, it is very possible that the low dependence of the clock on light is because the compensation by temperature is lost. Without a temperature compensation the authors can only stress that the light input is low to the clock but not that the clock regulation of gene expression is low. This must be altered throughout the text.

Response: We agree with the reviewer's point that temperature has influence on the circadian clock and control diurnal gene expression. We preferred to exclude this variable from the equation in order to study the responses to light/dark transition. We emphasize this point in the conclusion with:

'The exception to this rule is spruce which shows a largely arrhythmic expression. This suggests that despite its universality and ancient origin, diurnal gene expression can be suppressed to better suit a particular environment. However, we cannot exclude that in combination with the diurnal temperature oscillations, the amount of rhythmic genes would increase in spruce.'

3. Discussion in lines 238-253 are very speculative. When comparing the circadian expression of two different species grown on different light regimes, the differences referred by the authors may reflect exactly these differences in day-light regimes, and not a particular characteristic of the species' circadian clock. Without a control in which both species are grown and sampled in the same experimental frame of light, photoperiod and temperature, only predictions and assumptions, can be made.

Response: We agree with the reviewer that differences in the photoperiod between the experiments could cause problems with our interpretations. However, we argue that this is not an issue, because despite the differences in the photoperiod, the diurnal transcriptomes of *C. paradoxa* (16L/8D) and *C. reinhardtii* are still significantly similar. Overall, we found that *C. paradoxa* genes tend to peak 2 hours earlier, which does not reflect the 4 hours difference in the photoperiod.

We also investigated how the differences in photoperiod influence diurnal gene expression, by comparing two *A. thaliana* studies that used diverse photoperiods in seedlings and rosettes (please see the new Figure S7 below). We found that comparing 16L/8D to 12L/12D causes a modest shift difference of 0.52 hours, while comparing 8L/16D to 12L/12D causes a modest shift of -0.9 hours. Interestingly, we observed the largest differences when comparing data from the two studies. For example, 12L/12D (Michael et al., 2010) and 12L/12D (Blasing et al., 2005) caused a shift of 2.89 hours. However, whether this is caused by the developmental stage (seedlings vs. rosettes) or temperature (hot/cold vs. hot/hot) is unclear.

To summarize, we stand by our point that the diurnal transcriptomes are similar, as the used photoperiod differences (16L/8D, 12L/12D, 8L/16D) have minor influence on the phase of the gene expression in *A. thaliana*.

4. While I tend to agree with authors that Charophyceae developed a special gene expression mechanisms adapted to multicellularity, the species used in this experimental approach, is only filamentous and single-celled. Without circadian data from a real “talophytic” algae, such as the recently sequence “Chara braunii” (Nishiyama et al, 2018, Cell vol 174) this would be very speculative. I would rather “suggest” and insist in this lack of data.

Response: We agree with the reviewer that additional Charophyceae species, such as Chara, would be a valuable addition to the dataset. We have added this sentence to the discussion to highlight the point:

‘However, additional diurnal expression data from other Charophyceae, such as the recently sequenced Chara braunii (Nishiyama et al, 2018), is needed to support this observation.’

5. In microalgae, ADAGIO-LIKE genes such as ZTL or FKF1 do not show a complete domain structure, rather the Fbox, Kelch repeat and LOV domains are separated in two or three proteins. It is curious that a rather ancient clade such as the Glaucophytes would have so clear a ZTL orthologous. Authors should discuss on this remarkable discovery or check the data and see if the protein structure is like a true ZTL protein.

Response: We thank the reviewer for bringing this up. The closer inspection of the protein sequence alignment of the ZTL gene family revealed that the gene from Glaucophytes was incorrectly assigned to it by OrthoFinder. To remedy this, we have constructed new families based on the literature and manual inspection of the sequence alignments (please see Supplementary Figure 16 to view the alignments). As suggested by the reviewer, Glaucophytes do not have representatives in this family. This has been amended for the other clock families and the text.

Minor points

- References for synchronous cultures in algae could be more up to date than the 1979 ref.

Response: We have updated the citation to Hlavová M. et al., 2016.

- The concept of “old” genes should be better explained by authors, as genes themselves have no age, are rather the result of longer or shorter evolutionary processes in any particular species and is difficult to generalize.

Response: We agree that this section could use clarification, and we have modified the sentence to:

'This analysis revealed that the majority of genes ~~are old~~ belong to the earliest phylostratum for algae and plants (Fig. 2A, green bar), suggesting that most genes in plants have ancient origin'.

- In several figures (such as in 2B) please define the X-axis scale and show the scale independently in every figure, not just in supplemental files.

Response: We thank the reviewer for pointing this out. Figure 2B and others have been amended, where necessary.

- It would be informative to indicate in Figure 3E which percentage of the analysis defers from the diagonal distribution, which is the statistical approach of this significance and which percentage of genes are diagonally distributed in each species. A look at the difference figures shows that actually most of the genes are sited out of this diagonal distribution.

Response: We agree that this would be informative and we added Supplementary Table 13, which shows the percentages of genes found in the diagonal, before and after the phase shift.

- Line 826 please correct “indicates”.

Response: Corrected.

- Except when deliberately made a common name (and then define first time mentioned), please use genera and species names in italics.

Response: Corrected throughout the manuscript.

- Line 286 and thereon. In this discussion the word “ubiquitous” is not well understood. Are authors referring to space ubiquity or to ubiquitous expression in time. Or rather, it is being used in a rather subjective way referred to the way the data in the figure is organized?

Response: We agree with the reviewer that ‘ubiquitous’ is unclear. We refer to ubiquitous expression in time. To make this more clear, we rewrote ‘ubiquitous’ to ‘uniform’ throughout the manuscript. For example: *'Conversely, the more complex multicellular land plants show ~~ubiquitous~~ uniform expression of these genes throughout the day (Fig. 1D).'*

- Supplemental Fig 6. Should not the colored panels on the right have a 24h time scale?

Response: We agree with the reviewer that this might be unclear. We have updated the figures containing these heatmaps with numbers indicating the time, 0-24.

- Regarding one of the last conclusions, the identification of CCA1/LHY orthologous proteins in Cyanophora and Porphyridium is quite significant. Have you tried to use these proteins and search in cyanobacteria databases to see if the genes precede in some cyanobacteria divisions the symbiotic event? This would be a very interesting point in the clock evolution.

Response: We performed the search, but we could not find any significant BLAST hits against cyanobacteria *Synechocystis*, *Nostoc*, *Synechococcus* or *Prochlorococcus*. It seems that these genes are specific to Archaeplastida.

- Fig 5. Please explain and show x-scale on the right panels.

Response: We have updated the figures containing these heatmaps with numbers indicating the time, 0-24.

Reviewer #3 (Remarks to the Author):

In this manuscript the authors present a comparative analysis of the dynamics of gene expression under light/dark cycles in different species from different Archaeplastida lineages and cyanobacteria. They use publicly available transcriptome data from a cyanobacterium (*Synechocystis* sp. PCC6803), a chlorophyte (*Chlamydomonas reinhardtii*) and land plants, (*Arabidopsis thaliana*, *Oryza sativa*) and generate transcriptome data for a glaucophyte (*Cyanophora paradoxa*), a rhodophyte (*Porphyridium pupureum*), a charophyte (*Klebsormidium nitens*) and a gymnosperm (*Picea abies*). They observe significant conservation of the phases of gene expression among orthologs, complementing single species studies in algae (Noordally and Millar 2015) and land plants (Filichkin et al., 2011, PMID: PMC3111414). This study also demonstrates that older genes are more likely to cycle than newer lineage specific genes. The authors also provide analysis of the orthologs of land plant clock genes. The authors conclude that there is a loss of diel control of cell cycle in multicellular species and general decreased diel control of gene expression in multicellular organisms.

The authors generate an interesting and valuable dataset for comparative analyses of diel expression in photosynthetic organisms. My main concern is their conclusion on loss diel regulation of the cell cycle and overall gene expression in multicellular organisms.

Response: We appreciate the thorough analysis and the great suggestions that helped us to improve the manuscript.

They observe that the percentage of cell cycling genes that had diel oscillations was lower in the multicellular organisms. The tissue sampled from land plants probably contained very few dividing cells since they were, I believe, from mature leaves (although the plants were actively growing). This is likely very different from unicellular algae, in which, depending on the light intensity and nutrient status a significant percentage of cells will be dividing at a given cycle. It has been recently shown that in young *Arabidopsis* seedlings DNA replication is regulated by the circadian clock (Fung-Uceda et al., PMID:29576425). In addition, meristems in *Arabidopsis* appear to have more robust rhythms than other tissues (Takahashi et al., PMID:26406375).

Response: We thank the reviewer for pointing out these excellent papers. We now mention them in the discussion when addressing the decreased rhythmicity in land plants.

‘Indeed, it has been observed that actively dividing cells in the shoot apex show more robust rhythmicity than roots, indicating that different tissues can be under alternative diurnal regulation (Takahashi et al., 2015).’

and

‘While DNA replication in young seedlings is regulated by the clock (Fung-Uceda et al., 2018), cells in the multicellular organisms showed a more uniform expression of cell division markers (Figure 1D), suggesting that cell division in mature tissues differs from young tissues.’

Therefore, for a proper comparison actively dividing cells from land plants would need to be compared to actively dividing cells from unicellular algae. The apparent lower rhythmicity of *Klebsormidium*, speaks for a role of multicellularity per se, since this charophyte lacks differentiated cells. However, this observation is on only one species measured under one condition, which is not sufficient prove for such a strong and important conclusion. In the absence of experimental studies on the synchronization of cell division in the species studied, the authors should provide a more nuanced interpretation of this result in the abstract and at the end of the introduction, clarifying the different alternative hypothesis.

Response: We agree with the reviewer that comparison of actively dividing cells would be an interesting study. However, we claim that the majority of the cells in complex plants are not dividing rhythmically, which is based on the observation shown on Figure 1D. This indicates that, unlike in algae, most of the sampled cells in the multicellular species are largely uncoupled from the diurnal cycle.

We agree that this might not be the case in actively dividing cells, such as meristems. We emphasize this with:

‘While DNA replication in young seedlings is regulated by the clock (Fung-Uceda et al., 2018), cells in the multicellular organisms showed a more uniform expression of cell division markers (Figure 1D), suggesting that cell division in mature tissues differs from young tissues.’

And the final sentence of the conclusion:

‘To further explore if the conservation of diurnal regulation is found across different tissue types, developmental stages, as well as protein and metabolite levels will be an interesting and promising area of future investigation.’

The authors also observe that the phases of gene expression in multicellular organisms are more widely distributed than in single celled algae. They conclude that this result indicates a less "defined diurnal control". However, it could also be interpreted as a more sophisticated fine-tuned diurnal control in which time can be measured and defined with more precision. This might be necessary to adjust the needs of different cell types. Although the circadian regulatory network in green algae is not well characterized the increase in the number of circadian clock components in land plants (as the authors show) suggests a need of a more complex regulation of time dependent expression. It is also important to take into consideration that the phase distribution of gene expression varies under different growth conditions (Michael et al., PMID: PMC2222925) and based on the developmental stage of the organ, for example older leaves in rice have a wider more monophasic distribution of

phases than young seedling leaves (Xu et al., 2011). It would be interesting to see how diel transcriptomic datasets from *Arabidopsis* seedlings (such the ones from Michael et al., 2008) differ from mature leaves. Here again a more nuanced interpretation of the results is necessary and additional analyses might help clarify this point.

Response: We thank the reviewer for the great suggestion to include this analysis, which is shown below:

Supplementary Fig. 7. Comparison analysis of diurnal expression of different developmental stages in *O. sativa* and *A. thaliana*. The heatmaps indicate the phase comparisons of genes in the different datasets. The color intensity of the cells indicates the number of genes that peak at a given phase combination in the two datasets. Black lines indicate a transition from light to dark. The percentages indicate the rhythmic genes identified in the different datasets. The histograms show the relative frequency of phase differences between genes in the different datasets. The upper number indicates genes which were identified as rhythmic in both datasets while the lower number indicates the average Δ phase. A) Four different datasets

were compared for *A. thaliana* 12L/12D, 22°C/12°C; 16L/8D; 8L/16D; 12L/12D 22°C/22°C with two and three replicates. B) Two different datasets were compared for *O. sativa*; seedlings and flag leaves.

We have added the following paragraph to the Results section:

'To better understand how developmental stages and the photoperiod variation influences the observed rhythmicity, we compared diurnal data from A. thaliana seedlings (12L/12D, 22°C day/12°C night and constant 22°C, in 16L/8D and 8L/16D)46 and rosettes (constant temperature, 12L/12D)32 (Supplementary Fig. 7A). We observed that rosettes contain a higher percentage of rhythmic genes (37%) than any of the seedling experiments (10.2%, 17% and 26.4%, Supplementary Fig. 7A). This was also observed in O. sativa, where seedlings (33.2%) showed lower percentage of rhythmic genes than flag leaves (43.5%)(Supplementary Fig. 7B)31. While the diagonal distribution of the shift values indicated that the diurnal transcriptomes are highly similar across the photoperiods, developmental stages and laboratories (Supplemental Fig. 7A, B), we observed a substantial average difference between phases of A. thaliana rosettes and seedlings (rosettes peak 2.83, 2.98 and 4.96 hours earlier than 12L/12D, 16L/8D and 8L/12D seedlings). These results indicate that while the genes tend to robustly peak in the same sequence across different conditions, the overall phase of the genes can be substantially influenced by the environment and developmental conditions.'

In general, the results require a deeper and more nuanced interpretation and discussion. Below are some suggestions.

Other comments:

1. It would be interesting to know which are the genes (orthologs) with highly conserved expression patterns. Are there any processes with more of those genes? That will help understand which processes require cyclic gene expression.

Response: We have performed the suggested analysis by identifying orthologues that peak within ± 2 hours. The result is shown on Figure S6:

Supplementary Fig. 6. Similarity of the phases of the rhythmic genes in the different biological processes and species. The heatmap shows the fraction of rhythmic orthologs assigned to a specific biological process, that peak within ± 2 hours of each other. For example, value of 0.6 would indicate that 60% of orthologs assigned to a given process peak at similar time during the day. The organisms are grouped by organismal complexity: single cellular-single cellular (S-S), single cellular-multicellular (S-M) and multicellular-multicellular (M-M). The last three columns indicate the average expression of the three groups. Biological processes are sorted from the most to the least abundant.

Based on the figure, we now state in the manuscript:

‘To understand which biological processes show conserved rhythmic expression across Archaeplastidia, we calculated the fraction of the orthologs. We observed a conservation of rhythmicity of signalling, tetrapyrrole synthesis, cell division, protein, RNA and DNA synthesis and processing pathways (Supplementary Fig. 6). Conversely, gluconeogenesis, oxidative pentose pathway, tricarboxylic acid cycle and amino acid metabolism were among the less rhythmic pathways, suggesting that diurnal regulation of primary metabolic pathways is less conserved (Supplementary Fig. 6).’

2. Line 140: The authors use "lag" values to describe the phase of expression. Phase is a more commonly used term. Phase should be represented as zeitgeber time (ZT) which is "hours after dawn" a metric commonly used in chronobiology. The "lag" values appear at first to be defined as time point of the series, ie. it initially appears that lag 3 would not be the same in a 2 h resolution experiment than at 4 h resolution experiment. The use of "phase" and ZT would be less ambiguous.

Response: We thank the reviewer for the valuable suggestion. We have replaced ‘lag’ with ‘phase’ throughout the manuscript to be more precise.

2. Line 151/Line 380: The authors observe that *P. abies* have significant lower levels of cyclic gene expression. The authors hypothesize that this is due to its origin at very high latitudes and therefore exposure to extreme changes in photoperiod. The

interpretation/discussion of this result needs to be improved and the following points addressed:

a. This argument is often use on the opposite way, such that exposure to changes in photoperiod require the presence of a strong circadian oscillator. Is there any evidence for weaker diel rhythms in species originating from northern latitudes?

Response: This is a very good point. We have compared other studies of gymnosperms, which suggested that there is no connection between the low rhythmicity, gymnosperm lineage or northern latitudes. We have added this text to the results section to highlight this:

'The basis of the weak diurnal rhythms observed in spruce (63°N) is unclear, as another gymnosperm, Douglas-fir (42-43°N), shows substantially higher diurnal rhythms (29%, Cronn et al., 2017). Conversely, Japanese cedar (31-41°N) shows comparably weak diurnal rhythms (7.7%, Nose and Watanabe, 2014). These results indicate that the weak diurnal rhythms in spruce are not general for the gymnosperm lineage, and are not linked to the northern latitudes.'

b. There are many *P. abies* genotypes and some are surely adapted to more southern regions. It would be important to know the ecotype used.

Response: We added the genotypes and their origin. The ecotypes are *Synechocystis* sp. PCC 6803, *Cyanophora paradoxa* UTEX555 (SAG 29.80, CCMP329), *Porphyridium purpureum* SAG 1380-1d, *Klebsormidium nitens* NIES-2285, *Physcomitrella patens* Gransden wild type, *Selaginella moellendorffii* (clone of the sequenced *S. moellendorffii*), *Picea abies* offspring of the Z4006 clone (sampled from Ragunda, central Sweden), *Oryza sativa* (subsp. *japonica* var. *Nipponbare*) and *Arabidopsis thaliana* (Colombia-0).

d. Gyllenstrand et al., 2014 (PMID:24363286) showed that in *P. abies* (L. Karst) the amplitude of diel expression of some circadian related genes is significant higher under short days than under long days. Maybe photoperiod has a general influence of rhythmicity in spruce.

Response: We have been informed by Ove Nilsson (Umeaa) that subjecting spruce to 12L/12D photoperiod (or shorter) could cause the seedlings to enter hibernation state, which could shut down the diurnal gene expression. The minimum recommended photoperiod was 16L/8D. However, we agree with the reviewer that photoperiod has a large influence on the rhythmicity, and we have added this sentence to the results section:

'Indeed, shorter photoperiods in spruce (8L/16D) are accompanied by an increase of amplitude of clock genes (Gyllenstrand et al. 2014).'

On Line 353 the authors write "suggesting that the clock is running differently or is not active in spruce". What do you mean "differently"?

Response: We have modified the sentence to: *'Surprisingly, all of the clock components are present in spruce (Fig. 5C-I) with the exception of ELF3, suggesting that the spruce clock might be more active at different developmental stages, temperature oscillations or photoperiods than were used in this study.'*

As shown by Gyllenstrand it is possible that a different photoperiod or other condition (temperature, light intensity) leads to stronger rhythms.

Response: We have added the publication to the text: *'Indeed, shorter photoperiods in spruce (8L/16D) are accompanied by an increase of amplitude of clock genes (Gyllenstrand et al. 2014).'*

c. There are other studies on diel transcriptome expression in gymnosperms. Nose & Watanabe (2014) (PMID: 25403374) analyzed the diel expression in Japanese cedar and concluded that about ~7% of the genes cycled. Cronn et al., (2017) analyzed diel expression in Douglas-fir needles and showed that ~30% of genes cycles. That is similar to another tree, poplar (Filichkin et al., 2011). These other experiments should be

Response: We thank the reviewer for pointing out these useful publications, which we have used to improve the text (please see the comment above).

3. Line 161: what do you mean with double peaks.

Response: We have changed 'double peaks' to 'bimodal distribution'.

4. Line 193: The authors observe that there is a significant tendency for genes from older phylostrata to have higher rhythmicity than genes from genes from younger phylostrata. This agrees with a study using chlamydomonas that demonstrated that there is a significant tendency of cyclic gene expression to be maintained after gene duplication (Panchy et al., PMID: 25354782).

Response: We have added this sentence to the paragraph:

'Since rhythmicity tends to be retained after gene duplication (Panchy et al., 2014), we hypothesize that the high rhythmicity of the old phylostrata has ancient origin.'

5. Line 207 " We therefore conclude that older phylostrata tend to be more strongly expressed". Is that because these genes are expressed in more tissues?

Response: The reviewer raises a good point. However, since we have observed this phenomenon in single-cellular organisms, we do not think that this is due to more ubiquitous expression. To make this clearer, we have modified the sentence:

'Since we have observed these trends in single-cellular organisms, we hypothesize that genes belonging to older phylostrata are more strongly expressed in all cell types.'

6. Line 232: The DeltaLag description is confusing because it is unclear when a distribution or the average is described. Does this sentence (on line 232) mean:

"The analysis revealed that, in contrast to evenly distributed DeltaLagExpected values with an average DeltaLagExpected of 5.98 the distribution of DeltaLabObserved values peaked at +2 hours and the average DeltaLag Observed was 4.07."?

Response: To improve the clarity of this section, we have removed the average deltaLag (now Δ phase, as we have renamed 'lag' to 'phase'), as the value is not needed to understand this rather difficult part of the manuscript. The simplified sentence is:

'Indeed, the analysis revealed that, in contrast to evenly distributed Δ phase_{expected} values, Δ phase_{observed} values peaked at 2-3 hours (Fig. 3B).'

7. The Chlamydomonas original experiment had 1 h resolution sampling, but on line 447 the authors indicate that they use samples taken every 2 h. I assume that this is to match better the other studies, but an explanation is needed.

Response: The reviewer is correct. We have added this sentence to clarify this statement:

'Only samples taken every two hours were considered for the analysis to better match the sample density of the other species.'

Related to sampling frequency, the authors should mention that an increase sampling density increases the number of genes classified as cycling using JTK cycle (Li et al., 2015;

PMID:25662464) and they should discuss whether the differences in their experiments could influence their results about the percentage of cycling genes.

Response: The reviewer is correct. Supplemental Fig. 14 shows that when sampling density is decreased from every 2 hours, to every 4 hours, the percentage of rhythmic genes drops on average by 26%. We mention this in the methods:

'We observed an average decrease of 26% of rhythmic genes when sampling density was decrease from 12 daily time points to 6 daily time points, indicating that the amount of rhythmic genes is most likely higher than we report.'

8. Line 272: "with an anticipatory decrease below 50% of maximum expression before dusk", Since the expression of FBPsases and RPE increase just before dusk, I would change this description to "with a decrease below 50% of maximum expression during the second half of the light period".

Response: The sentence is amended.

9. Line 316: It has been demonstrated that in Arabidopsis CCA1 and LHY repress the expression of the PRRs (Kamioka et al., 2016; PMID: 26941090). That needs to be corrected on Figure 5A as well.

Response: The figure is amended.

10. Line 322: It is misleading to say that GI and ZTL repress the expression of TOC1 and other PRRs genes. GI and ZTL are involved in the degradation of TOC1 and PRR5 proteins, not in the transcription of the respective genes.

Response: We thank the reviewer for pointing this out. We now indicate that GI and ZTL are causing degradation of these proteins.

11. Line 333: I would not call it an "incomplete" Viridiplantae clock, I would say "despite of the lack of conservation of the Viridiplantae clock". Since it is possible that these algae have a circadian clock as the authors indicate at the end of that sentence.

Response: The sentence is amended.

12. Figure 5B1: TOC1 proteins should be included in the PRR tree. TOC1s are PRR proteins and in the green algae "TOC1s" cannot be assigned into either "TOC1" or "other PRRs" (Linde et al., 2017; Liu et al., 2013 PMID:23856081). The description of the PRRs in the text should include TOC1 and other PRRs, trying to define a Chlamyomonas TOC1 is misleading.

Response: We agree with the reviewer. TOC1 and PRRs were separated by OrthoFinder. To remedy this, we are now basing the clock families on manually curated alignments and literature. TOC1 and PRR proteins now constitute one family, as expected, which is shown in the amended figure.

13. Figure 5H: why are there two TOC1 genes in Arabidopsis?

Response: Similarly to the comment above, this is an OrthoFinder error, which is now corrected.

14. Sup. Fig. 7 and 8 trees should be combined in one.

Response: We have fused the trees as per reviewer's suggestion and the literature search.

15. It would have been good to include *Ostreococcus tauri* data because in this green alga the function of "conserved" clock components CCA1/LHY/RVE and PRR have been described.

Although clock components have been identified in *Chlamydomonas* the clock network is poorly characterized.

Response: We fully agree with the reviewer and have included *O. tauri* in the clock analysis. Interestingly, the expression patterns of *O. tauri* and *C. reinhardtii* are similar, which we comment on in the text.

16. Line 358: "the lower specific expression pattern", do you mean the broader peaks of expression? Sequencing data not only provides relative expression abundances, it allows to determine the overall expression level of genes relative to others, is the expression of clock genes in spruce lower than in other plants?

Response: We are not confident that the expression values across species can be compared in this manner, as RNAseq data provides relative expression abundances across species. For example, let us assume that we have two species (A and B) which contain gene *a*, expressed as 3 mRNA molecules in both species. However, species B also contains gene *b*, also expressed as 3 mRNA molecules.

Let us assume that we sequenced all mRNA molecules for both species. After normalizing for the sequencing depth (3 molecules in species A, 6 in species B), the expression of gene *a* in both species differs, because the reads are 'taken' by gene *b* in species B. The data is thus relative across species (please see the above example). However, the data can be considered absolute within one species, and it is therefore feasible to compare expressions of gene *a* and *b* within species B. As spruce has more genes compared to *Arabidopsis*, we do not feel comfortable comparing and discussing gene expression across these plants.

17. Methods.

a. Which data set from rice was used? It seems that in the study from Xu et al., 2011 to types of leaf, flag leaf and seedling leaf were used. The both types of leaves appear to have slightly different phases of gene expression, with the seedling leaves displaying a more narrow range of phases.

Response: We are using expression data from seedling, which we now emphasize in the manuscript. The comparison of the seedlings and flag leaves is shown in Supplementary Fig. 7B:

The analysis shows that the phases of genes from seedlings and flag leaves are similar, as the genes form a diagonal on the phase plot, and the Δ phase plot shows values centred around 0.

b. The strain/ecotype information of all the species used needs to be included.

Response: As stated above, we have now provided all strain identifiers used in the study.

c. Line 456: why was there a different in "detection" threshold for Arabidopsis and Oryza, please include an explanation.

Response: This was caused by Arabidopsis and Oryza data having three and two duplicates, respectively. This is now explained as follows:

'For A. thaliana, MAS5 software (Pepper et al., 2007) had to call a gene 'present' in at least two out of three replicates, while for O. sativa, in at least one out of two replicates.'

d. The new transcriptome data should be deposited in a public database.

Response: We apologize for this omission. A link to EBI is now given:

Data availability

The TPM-normalized expression data are available in Supplementary Table 2-11, while the fastq files representing the raw RNA sequencing data are available from EBI Array Express under accession number E-MTAB-7188.'

18. Minor edits:

Figure 1C and D, font too small

Response: The fonts are increased.

Figure 2 fonts too small

Response: The fonts are increased.

Figure 3 the smaller fonts are too small

Response: The fonts are increased.

Figure 5A: what do the colored lines in the gene symbols mean?

Response: We have explained the lines in the figure legend:

'The colored bars within the gene families indicate in which plant clades a given gene family is present.'

Figure 5 B tree, font too small

Response: The fonts are increased.

Line 659: in this reference, check name of author

Response: The name has been corrected.

References: several references lack page numbers

Response: We have corrected the references.

We thank the reviewer again for helping us improve the manuscript!

Reviewers' comments:

Reviewer #1 (Remarks to the Author):

I remain quite enthusiastic about this manuscript. I think the responses to the first set of reviews have improved the manuscript. However, I also think that some of the response have created some new issues that need to be addressed. Nonetheless, I do not feel that any of these concerns require new analyses—I think they can be addressed textually.

This work offers three important insights:

1. Diurnal transcriptomes are surprisingly similar across] evolutionary distance, and a range of morphologies and habitat. Spruce, however, is an exception.
2. The age of a gene family is positively correlated with both the level of rhythmicity and the expression level, but does not influence the diurnal expression pattern. Expression level and expression pattern are subject to distinct control mechanisms.
3. The loss of clock-controlled cell division was associated with the origin of multicellularity, rather than with the emergence of land plants. This is an important conclusion. However, as raised in the reviews, this conclusion may be weakened by the restriction of cell division to a small subset of cells in land plants.

The authors also note that the basal lineages appear to have simpler clocks missing some critical components, based on the Arabidopsis clock.

The authors have provided a large and important set of gene expression data. I think that this manuscript will provoke much deeper thinking on the evolution of clocks in general and specifically in the green lineage. I think that this work will stimulate future experimentation. I'd like to see this work published.

Major points

Line 356-371 I think the short description of the clock mechanism needs work. I would also point out that the positive regulation of the PRRs by CCA1 and LHY is probably indirect as to date CCA1 and LHY have been described as repressors (although their action as activators in other complexes has not been excluded). I would also add that providing a succinct and accurate description of the clock mechanism as described in Arabidopsis is nightmarishly difficult. So less is probably preferable to more, because the more you say the greater the likelihood that you will say something incorrect or incomplete.

Line 372-391 I do not see how the expression analyses addressed the evolution of either phototropism or photoperiodism First, phototropism is not addressed at all. Second, the authors are really discussing when the clock as understood in Arabidopsis evolved, not how it evolved.

Line 385 the authors suggest that the Viridiplantae might have an uncharacterized type of clock. I agree that they are missing a number of important components. However, recent work from the Harmer lab showed that one could eliminate both CCA1/LHY- and the RVE8-clade transcription factors yet retain a remarkably normal clock (PNAS 115:7147, 2018). So this "uncharacterized clock" might well be a simpler loop based on CCA1/LHY and LUX. Calling it "an uncharacterized clock" to me implies something wholly different. Certainly that cannot be excluded, but the actual solution might be simpler—a reconfiguration of known components. Note also that the authors (nor Stacey Harmer in her review ref 50 on which Fig 5A is based) have not considered other known Arabidopsis clock

components like LWD1 and LWD2 and their interacting TCP transcription factors (see Nat Commun 7:13181, 2016). Furthermore, in Arabidopsis, transcriptional activation of ELF4 is provided by FAR-RED ELONGATED HYPOCOTYL3 (FHY3), FAR-RED IMPAIRED RESPONSE1 (FAR1), and ELONGATED HYPOCOTYL5 (HY5), three transcription factors that are positive regulators of phytochrome A signaling (Nat Cell Biol 13:616, 2011). Although these factors are usually ignored in the reviews, they probably should not be. So I think there is a great deal more complexity to the Arabidopsis clock, especially in the form of transcriptional activators that have been ignored in the “repressilator” mindset, than the reviews typically discuss. That suggests to me that there may be additional clock components in the Viridiplantae that have not been considered. I am not suggesting that the authors do more analyses. I am simply suggesting that they might want to suggest that the “uncharacterized clock” could be formed through a reconfiguration of known clock components, perhaps with the recruitment of some new components. Perhaps it is worth considering that the Viridiplantae diurnal transcriptome has a fairly simple topography with one or two sharp phases of peak expression that could be controlled by a clock with simple architecture. With evolutionary complexity the phases of peak expression broaden. It is possible that the recruitment of additional components, like the PRRs, enabled this broader distribution of phases of peak expression. Reviewer 3 also brought up this issue.

Lines 327-342. This returns to the idea raised by Reviewer #3, of diurnal regulation of cell division genes. I think that the authors should point out that some of this apparent loss of rhythmicity of cell division genes with the acquisition of multicellularity might be attributed to the relegation of division to a small number of meristematic cells, diluting the expression patterns of cell division genes in a majority population of non-dividing cells. That would be consistent with their point that it was multicellularity rather than colonization of land that drove this change. This issue recurs in lines 457-460

Lines 415-420 Spruce is puzzling. Although Douglas fir is more rhythmic than spruce, latitudes of 43 are not really as extreme as latitudes of 63°N. However, Arabidopsis accessions from latitudes of 60° or greater are robustly rhythmic (by leaf movement—see Science 302:1049, 2003). So I do not think that the authors should not feel badly that they have not resolved the spruce puzzle. But also, the argument that it is extreme north latitudes that are responsible for the loss of rhythmicity is tantalizing but probably not true in any general sense. So I think the authors should acknowledge that spruce is an outlier. I think they should be cautious in trying to explain why it is an outlier. The answer in the short term may simply have to remain “because.”

Minor points

Line 40 innumerable means “too many to be counted.” Although it is often used hyperbolically, would it not be better to be restrained and simply say “many?”

Lines 105-106. ... despite the large evolutionary distances and differences in morphological complexity and habitat among the sampled taxa.

Line 109 ... a broader phase of peak expression...

Line 128 where were

Line 142 ... we used the JTK_Cycle algorithm...

Line 144 ... phase of peak expression...

Line 145 wouldn't the 12th time point be at ZT 23, not 24?

Line 165-167 perhaps gene expression in angiosperms is less under the control of the diurnal cycle, but might it not be preferable to add the possibility that it might be under more nuanced regulatory control allowing access to more phase bins. Perhaps that is some disadvantage to sharply peaking expression patterns as seen in algae and early-diverging land plants.

Lines 168-178. This paragraph discusses the apparent loss of diurnal regulation of cell division genes. Might not at least some of this apparent loss be attributed to the relegation of division to a small number of meristematic cells, diluting the expression patterns of cell division genes in a majority population of non-dividing cells?

Lines 198-199 There was a significant tendency for older phylostrata to have a higher proportion of rhythmic genes

Lines 215-217. Is this conclusion valid? Lacking cell type specific data, can one conclude that there is more expression in all cell types as opposed to very strong expression in a subset of cell types? I don't think so.

Line 232 4 hour difference

Line 248 and elsewhere. The authors speak of phases being delayed (or advanced). In the clock field those terms are generally used to discuss the alteration in phase (phase shift) following some stimulus. Might it be better to say that the phases are "offset" by two hours?

Line 257-258 another example-- ... if the phase of *C. paradoxa* genes were shifted offset by two hours in the future later in the diurnal cycle

Line 285 ... the fraction of orthologs that were rhythmic

Line 296 rosettes contain a higher percentage of rhythmic genes than seedlings in any of the seedling experiments.

Line 304 12L/12D, 16L/8D, and 8L/126D seedlings.

Lines 300-307 I agree that there are substantial phase difference among *Arabidopsis* rosettes and seedlings, but I think it is worth noting that the phase differences between rice seedlings and flag leaves is quite small.

Line 331 'cell.division' child bin??????? I suspect a typo, but if not could it please be more fully explained.

Line 396 whereby whereas

Reviewer #2 (Remarks to the Author):

Authors have answered most of the questions raised. I still am not very confident on the assumption in line 326 "...the expression of photosynthesis genes is not under diurnal control". It is difficult to accept that a plant would not regulate its photosynthesis gene expression diurnally, I would rather suggest "circadian" or better "light-circadian" dependence. Figures, tables and suggestions have been corrected.

Federico Valverde

Reviewer #3 (Remarks to the Author):

The authors have addressed most of my concerns adequately. However, although the authors provide an improved and more nuanced interpretation of their analyses in the result and discussion sections, the authors still write unsupported strong conclusions in the abstract and in the summary statement of the introduction.

In my opinion the authors conclusion that the cell cycle is not diurnally regulated in multicellular plants is not proved by their data. The authors observe strong oscillations in the expression of genes involved in cell division in the unicellular organisms they studied (Figure 1D). This result does not prove that cell division is diel regulated in these organisms, this was proven by others experimentally.

In the multicellular organisms studied, cell cycle genes display no or weak oscillations. Lack of diurnal control of cell division, which means cells would randomly divide independently of the time of day is just one of several hypothesis that can explain these gene expression patterns. In the response to the reviewers, the authors state that "the majority of cells in complex plants are not dividing rhythmically", which is true, however they should first state that the majority of the cells are probably not dividing at all (rhythmically or otherwise) in the material (leaf) they used for the analyses. The lack of cell division in most of the cells analyzed is an equally valid hypothesis that can explain their observations. This does not mean that cell division (in cells that actually divide) is not diel controlled in multicellular plants. If the authors want to emphasize the point of cell division in the abstract they should clearly state the different hypotheses that can explain their results and not select the preferred one.

In this version of the manuscript the authors also still claim (line 110) that multicellularity led to "less defined diurnal control" . This is a very vague statement that seems to mean that there is less diel control of gene expression in multicellular plants. Weak diel control is, again, only one hypothesis that explains their results. Different cell types in multicellular organisms display differences in the level and the phase of gene expression, this has been shown in plants (PMID: 2536376) and animals (for example: 29746605). Differences in the timing of gene expression in different tissues can lead to broader "peaks" of expression if tissues are not analyzed separately. Proving this second hypothesis is outside the scope of this manuscript, however, the authors should provide their readers with several hypothesis that can explain their results. The absence of a single clear hypothesis does not minimize the impact of this study. The manuscript should clearly state the important questions that remain to be addressed.

Reviewers' comments:

Reviewer #1 (Remarks to the Author):

I remain quite enthusiastic about this manuscript. I think the responses to the first set of reviews have improved the manuscript. However, I also think that some of the response have created some new issues that need to be addressed. Nonetheless, I do not feel that any of these concerns require new analyses—I think they can be addressed textually.

This work offers three important insights:

1. Diurnal transcriptomes are surprisingly similar across] evolutionary distance, and a range of morphologies and habitat. Spruce, however, is an exception.
2. The age of a gene family is positively correlated with both the level of rhythmicity and the expression level, but does not influence the diurnal expression pattern. Expression level and expression pattern are subject to distinct control mechanisms.
3. The loss of clock-controlled cell division was associated with the origin of multicellularity, rather than with the emergence of land plants. This is an important conclusion. However, as raised in the reviews, this conclusion may be weakened by the restriction of cell division to a small subset of cells in land plants.

Response: Based on the comments of Reviewer 1 and 3, we have addressed this issue in the current revision. Please see more detailed response in Reviewer 3's comments.

The authors also note that the basal lineages appear to have simpler clocks missing some critical components, based on the Arabidopsis clock.

The authors have provided a large and important set of gene expression data. I think that this manuscript will provoke much deeper thinking on the evolution of clocks in general and specifically in the green lineage. I think that this work will stimulate future experimentation. I'd like to see this work published.

Response: We thank the reviewer for showing appreciation of our work and for the new valuable comments and suggestions.

Major points

Line 356-371 I think the short description of the clock mechanism needs work. I would also point out that the positive regulation of the PRRs by CCA1 and LHY is probably indirect as to date CCA1 and LHY have been described as repressors (although their action as activators in other complexes has not been excluded). I would also add that providing a succinct and accurate description of the clock mechanism as described in Arabidopsis is nightmarishly difficult. So less is probably preferable to more, because the more you say the greater the likelihood that you will say something incorrect or incomplete.

Response: We agree with the reviewer that providing a brief yet accurate description of clock mechanism is rather challenging, but needed. Our aim is to give a very simplified description of the three main feedback loops embedded in the Arabidopsis clock. To emphasize this we added:

“The core of the clock is composed of three ~~3-major~~, integrated feedback loops (Fig. 5A) whose components are fine-tuned by additional regulatory factors.”

We now mention that the positive regulation by CCA1/LHY is probably indirect:

'It is important to note that the positive regulation of PRR7 and 9 by CCA1/LHY is probably indirect, as CCA1/LHY are shown to be repressors⁵⁵, suggesting that these two genes are involved in a yet uncharacterized regulatory circuit.'

Line 372-391 I do not see how the expression analyses addressed the evolution of either phototropism or photoperiodism First, phototropism is not addressed at all. Second, the authors are really discussing when the clock as understood in Arabidopsis evolved, not how it evolved.

Response: We agree with the reviewer and we modified the sentence to:

"To explore how the clock as observed in Arabidopsis evolved, we analysed the expression of eight gene families containing the core clock genes from A. thaliana (CCA1, LHY, RVE1-8, PRR3, 5, 7, 9, TOC1, LUX, ELF3, ELF4, GI and ZTL) in the nine members of the Archaeplastida (Fig. 5C-I)."

Line 385 the authors suggest that the Viridiplantae might have an uncharacterized type of clock. I agree that they are missing a number of important components. However, recent work from the Harmer lab showed that one could eliminate both CCA1/LHY- and the RVE8-clade transcription factors yet retain a remarkably normal clock (PNAS 115:7147, 2018). So this "uncharacterized clock" might well be a simpler loop based on CCA1/LHY and LUX. Calling it "an uncharacterized clock" to me implies something wholly different. Certainly that cannot be excluded, but the actual solution might be simpler—a reconfiguration of known components. Note also that the authors (nor Stacey Harmer in her review ref 50 on which Fig 5A is based) have not considered other known Arabidopsis clock components like LWD1 and LWD2 and their interacting TCP transcription factors (see Nat Commun 7:13181, 2016). Furthermore, in Arabidopsis, transcriptional activation of ELF4 is provided by FAR-RED ELONGATED HYPOCOTYL3 (FHY3), FAR-RED IMPAIRED RESPONSE1 (FAR1), and ELONGATED HYPOCOTYL5 (HY5), three transcription factors that are positive regulators of phytochrome A signaling (Nat Cell Biol 13:616, 2011). Although these factors are usually ignored in the reviews, they probably should not be. So I think there is a great deal more complexity to the Arabidopsis clock, especially in the form of transcriptional activators that have been ignored in the "repressilator" mindset, than the reviews typically discuss. That suggests to me that there may be additional clock components in the Viridiplantae that have not been considered. I am not suggesting that the authors do more analyses. I am simply suggesting that they might want to suggest that the "uncharacterized clock" could be formed through a reconfiguration of known clock components, perhaps with the recruitment of some new components. Perhaps it is worth considering that the Viridiplantae diurnal transcriptome has a fairly simple topography with one or two sharp phases of peak expression that could be controlled by a clock with simple architecture. With evolutionary complexity the phases of peak expression broaden. It is possible that the recruitment of additional components, like the PRRs, enabled this broader distribution of phases of peak expression. Reviewer 3 also brought up this issue.

Response: We thank the reviewer for the suggestion. We amended the text as follows:

"These results suggest that despite the lack of conservation of the Viridiplantae clock, conserved diurnal gene expression can be mediated by other mechanisms, such as light sensor-mediated response to light^{17,24}, carbon signalling^{19,60,61}, ~~or~~ an uncharacterized or a modified type of clock where the known components have been reconfigured to operate with other transcription factors involved in clock regulation⁶²⁻⁶⁴."

Lines 327-342. This returns to the idea raised by Reviewer #3, of diurnal regulation of cell division genes. I think that the authors should point out that some of this apparent loss of

rhythmicity of cell division genes with the acquisition of multicellularity might be attributed to the relegation of division to a small number of meristematic cells, diluting the expression patterns of cell division genes in a majority population of non-dividing cells. That would be consistent with their point that it was multicellularity rather than colonization of land that drove this change. This issue recurs in lines 457-460

Response: We thank the reviewer for raising this point up. We extended the text in order to explore the possibility of lack of rhythmicity because of lack of division as stated here:
Line 178: *“This suggests that either the increased morphological complexity of multicellular land plants necessitated uncoupling of the diurnal cycle and cell division, or that most cells in multicellular plants are not actively dividing, regardless of the diurnal cycle. The latter hypothesis is supported by the observation that the circadian clock controls cell division in actively dividing cells of *A. thaliana* seedlings⁴².”*

Other key points in the manuscript also discuss this point (please see our comments to Reviewer 3).

Lines 415-420 Spruce is puzzling. Although Douglas fir is more rhythmic than spruce, latitudes of 43 are not really as extreme as latitudes of 63°N. However, Arabidopsis accessions from latitudes of 60° or greater are robustly rhythmic (by leaf movement—see Science 302:1049, 2003). So I do not think that the authors should not feel badly that they have not resolved the spruce puzzle. But also, the argument that it is extreme north latitudes that are responsible for the loss of rhythmicity is tantalizing but probably not true in any general sense. So I think the authors should acknowledge that spruce is an outlier. I think they should be cautious in trying to explain why it is an outlier. The answer in the short term may simply have to remain “because.”

Response: We appreciate the reviewer suggestion. We emphasize the outlier behaviour of spruce by adding:

“These results indicate that the weak diurnal rhythms in spruce are not general for the gymnosperm lineage, and are not linked to the northern latitudes but are rather a characteristic specific to spruce.”

Minor points

Line 40 innumerable means “too many to be counted.” Although it is often used hyperbolically, would it not be better to be restrained and simply say “many?”

Response: The sentence has been changed from *“innumerable metabolic, physiological, and developmental responses to the external environment.”* to *“many metabolic, physiological, and developmental responses to the external environment.”*

Lines 105-106. ... despite the large evolutionary distances and differences in morphological complexity and habitat among the sampled taxa.

Response: The sentence is amended.

Line 109 ... a broader phase of peak expression...

Response: The sentence is amended.

Line 128 where were

Response: The sentence is amended.

Line 142 ... we used the JTK_Cycle algorithm...

Response: The sentence is amended.

Line 144 ... phase of peak expression...

Response: The sentence is amended.

Line 145 wouldn't the 12th time point be at ZT 23, not 24?

Response: JTK algorithm interpolates one phase between two collected time points. The first phase registered is the first time point collected and therefore, in a dataset where samples were collected every 2 hours for 24 hours, we will have 12 time points and 24 detected phases.

Line 165-167 perhaps gene expression in angiosperms is less under the control of the diurnal cycle, but might it not be preferable to add the possibility that it might be under more nuanced regulatory control allowing access to more phase bins. Perhaps that is some disadvantage to sharply peaking expression patterns as seen in algae and early-diverging land plants.

Response: We thank the reviewer for this great suggestion. We have added this sentence to this paragraph:

'Alternatively, the uniform diurnal gene expression in angiosperms suggests a more nuanced regulatory control allowing access to more phase bins.'

Lines 168-178. This paragraph discusses the apparent loss of diurnal regulation of cell division genes. Might not at least some of this apparent loss be attributed to the relegation of division to a small number of meristematic cells, diluting the expression patterns of cell division genes in a majority population of non-dividing cells?

Response: In line with the comments above, we have amended the sentence to:

Line 178: *"This suggests that either the increased morphological complexity of multicellular land plants necessitated uncoupling of the diurnal cycle and cell division, or that most cells in multicellular plants are not actively dividing, regardless of the diurnal cycle. The latter hypothesis is supported by the observation that the circadian clock controls cell division in actively dividing cells of *A. thaliana* seedlings⁴²."*

Lines 198-199 There was a significant tendency for older phylostrata to have a higher proportion of rhythmic genes

Response: The sentence is amended.

Lines 215-217. Is this conclusion valid? Lacking cell type specific data, can one conclude that there is more expression in all cell types as opposed to very strong expression in a subset of cell types? I don't think so.

Response: We hypothesized that genes belonging to older phylostrata are more strongly expressed in all cell types because we observed this behaviour in unicellular and monotypic multicellular organisms. However, the reviewer is correct that this is an assumption that we have not tested, and we have toned down the sentence to:

'Since we have observed these trends in single-cellular organisms, we propose that genes belonging to older phylostrata are more strongly expressed in all cell types.'

Line 232 4 hour difference

Response: The sentence is amended.

Line 248 and elsewhere. The authors speak of phases being delayed (or advanced). In the clock field those terms are generally used to discuss the alteration in phase (phase shift) following some stimulus. Might it be better to say that the phases are "offset" by two hours?

Response: We thank the reviewer for this suggestion. We have amended the manuscript to accommodate this.

Line 257-258 another example-- ... if the phase of *C. paradoxa* genes were shifted offset by two hours in the future later in the diurnal cycle

Response: We have amended the sentence to

*“if the phase of *C. paradoxa* genes were offset by two hours later in the diurnal cycle.”*

Line 285 ... the fraction of orthologs that were rhythmic

Response: The sentence is amended.

Line 296 rosettes contain a higher percentage of rhythmic genes than seedlings in any of the seedling experiments.

Response: The sentence is amended.

Line 304 12L/12D, 16L/8D, and 8L/126D seedlings.

Response: The sentence is amended.

Lines 300-307 I agree that there are substantial phase difference among *Arabidopsis* rosettes and seedlings, but I think it is worth noting that the phase differences between rice seedlings and flag leaves is quite small.

Response: We amended the text accordingly adding

*“Conversely, we observed only a modest difference of 0.5 hour between seedlings and flag leaves of *O. sativa*, suggesting that the developmental stage has a minor influence on diurnal gene expression in rice (Supplemental Fig. 7B).”*

Line 331 ‘cell.division’ child bin??????? I suspect a typo, but if not could it please be more fully explained.

Response: In MapMan gene ontology annotation, a child bin is a sub-bin of the main, and more general bin. In this example, ‘cell.division’ is a child bin of the more generic ‘cell’, which includes several child bins such as ‘cell.organization’, ‘cell.division’, ‘cell.cycle’.

To make this more clear, we have modified the sentence to:

‘In line with the observation that cell division is not similarly controlled in land plants (Fig. 1D), genes belonging to ‘cell’ bin (which contains genes involved in cell division) in multicellular organisms have a more uniform expression throughout the day, compared to the defined expression of unicellular plants (Fig. 4C, uniformly expressed genes indicated by dark green colour).’

Line 396 whereby whereas

Response: The sentence is amended.

Reviewer #2 (Remarks to the Author):

Authors have answered most of the questions raised. I still am not very confident on the assumption in line 326 "...the expression of photosynthesis genes is not under diurnal control". It is difficult to accept that a plant would not regulate its photosynthesis gene expression diurnally, I would rather suggest "circadian" or better "light-circadian" dependence.

Response: We amended the text as:

“This result indicates that in spruce, in contrast to C. paradoxa, the expression of photosynthesis genes is not ~~under diurnal control~~ light-circadian dependent.”

Figures, tables and suggestions have been corrected.

Federico Valverde

We thank the reviewer for the very helpful comments.

Reviewer #3 (Remarks to the Author):

The authors have addressed most of my concerns adequately. However, although the authors provide an improved and more nuanced interpretation of their analyses in the result and discussion sections, the authors still write unsupported strong conclusions in the abstract and in the summary statement of the introduction.

Response: We agree with the reviewer that other hypotheses should be mentioned as possible explanations for the reduced diurnal regulation in multicellular organisms. Therefore, we amended the text as follows:

Abstract: ‘However, in contrast to multicellular land plants, cell division and expression of most biological pathways in algae show specific expression depending on the time of the day.’

Introduction: “Conversely, land plants show a more uniform expression of cell division genes, suggesting that the cell division is either not taking place in most of the sampled cell types, or is not under diurnal control.”

Conclusion: “We found that diurnal programs are remarkably similar, despite the large phylogenetic distance spanning more than 1.5 billion years of evolution, ~~loss of light-controlled cell division in land plants (Fig. 1D)~~, and wide morphological diversity of the analysed species (Fig. 3E).”

In my opinion the authors' conclusion that the cell cycle is not diurnally regulated in multicellular plants is not proved by their data. The authors observe strong oscillations in the expression of genes involved in cell division in the unicellular organisms they studied (Figure 1D). This result does not prove that cell division is diel regulated in these organisms, this was proven by others experimentally.

Response: We agree with the reviewer, and as outlined above, we have either provided the alternative hypothesis (cell division is not taking place in the sampled organs) or removed this strong statement altogether.

In the multicellular organisms studied, cell cycle genes display no or weak oscillations. Lack of diurnal control of cell division, which means cells would randomly divide independently of the time of day is just one of several hypotheses that can explain these gene expression patterns. In the response to the reviewers, the authors state that “the majority of cells in complex plants are not dividing rhythmically”, which is true, however they should first state that the majority of the cells are probably not dividing at all (rhythmically or otherwise) in the material (leaf) they used for the analyses. The lack of cell division in most of the cells analyzed is an equally valid hypothesis that can explain their observations. This does not mean that cell division (in cells that actually divide) is not diel controlled in multicellular plants. If the authors want to emphasize the point of cell division in the abstract they should

clearly state the different hypotheses that can explain their results and not select the preferred one.

Response: We thank the reviewer for elaborating this point, which was not clear to us before. We fully agree with the possibility that cell division is not taking place at all in the sampled leaves. We have toned down the manuscript to rather focus on the fact that cell division and expression of biological processes show specific expression depending on the time of the day:

Abstract: 'However, in contrast to multicellular land plants, cell division and expression of most biological pathways in algae show specific expression depending on the time of the day.'

Introduction: 'Conversely, land plants show a more uniform expression of cell division genes, suggesting that the cell division is either not taking place in most of the sampled cell types, or is not under diurnal expression.'

Also, we have amended the results section to:

*'Conversely, the more complex multicellular land plants show uniform expression of these genes throughout the day (Fig. 1D). This suggests that either the increased morphological complexity of multicellular land plants necessitated uncoupling of the diurnal cycle and cell division, or that the lack of rhythmicity is a consequence of the lack of division, or only restricted to few types of cells, in the formed multicellular land plant. The latter hypothesis is supported by the observation that the circadian clock controls cell division in actively dividing *A. thaliana* seedlings⁴².'*

In this version of the manuscript the authors also still claim (line 110) that multicellularity led to “less defined diurnal control” . This is a very vague statement that seems to mean that there is less diel control of gene expression in multicellular plants. Weak diel control is, again, only one hypothesis that explains their results. Different cell types in multicellular organisms display differences in the level and the phase of gene expression, this has been shown in plants (PMID: 2536376) and animals (for example: 29746605). Differences in the timing of gene expression in different tissues can lead to broader “peaks” of expression if tissues are not analyzed separately. Proving this second hypothesis is outside the scope of this manuscript, however, the authors should provide their readers with several hypothesis that can explain their results. The absence of a single clear hypothesis does not minimize the impact of this study. The manuscript should clearly state the important questions that remain to be addressed.

Response: We thank the reviewer for raising this concern, which we agree on. We added to the text additional hypotheses:

'Establishment of multicellularity has also resulted in a broader phase of peak expression of most biological processes which, could be caused by a less defined diurnal control, or conversely, by a more elaborate diurnal regulatory control found in the different tissue types.'